# Optimization of Computational Intelligence Models for Landslide Susceptibility Evaluation

**Xia Zhao [1] and Wei Chen [1,2,*]**

[1] College of Geology and Environment, Xi'an University of Science and Technology, Xi'an 710054, China; 19209071022@stu.xust.edu.cn

[2] Key Laboratory of Coal Resources Exploration and Comprehensive Utilization, Ministry of Natural Resources, Xi'an 710021, China

\* Correspondence: chenwei0930@xust.edu.cn

**Abstract:** This paper focuses on landslide susceptibility prediction in Nanchuan, a high-risk landslide disaster area. The evidential belief function (EBF)-based function tree (FT), logistic regression (LR), and logistic model tree (LMT) were applied to Nanchuan District, China. Firstly, an inventory with 298 landslides was compiled and separated into two parts (70%: 209; 30%: 89) as training and validation datasets. Then, based on the EBF method, the *Bel* values of 16 conditioning factors related to landslide occurrence were calculated, and these *Bel* values were used as input data for building other models. The receiver operating characteristic (ROC) curve and the values of the area under the ROC curve (AUC) were used to evaluate and compare the prediction ability of the four models. All the models achieved good results and performed well. In particular, the LMT model had the best performance (0.847 and 0.765, obtained from the training and validation datasets, respectively). This paper also demonstrates the superiority of integration and optimization of models in landslide susceptibility evaluation. Finally, the best classification method was selected to draw landslide susceptibility maps, which may be helpful for government administrators and engineers to carry out land design and planning.

**Keywords:** evidential belief function; function tree; logistic regression; logistic model tree; Nanchuan District

## 1. Introduction

Landslides occur mainly due to the landform, geology, hydrology, soil, meteorology, human activities, and land-use patterns of the region under different geospatial and geographic conditions [1,2]. Landslides often represent a danger to human beings and cause property damage, due to natural conditions and human engineering activities [3]. In recent years, many areas of China have suffered from landslides, posing a clear threat to the safety of communication equipment, transmission lines, and transportation networks [4].

Landslide susceptibility assessment is an indispensable input into landslide risk management [5]. The selection of appropriate analysis, the scale of analysis, the choice of modeling methods, and the quality and quantity of available data in a landslide inventory map collectively determine the reliability of landslide susceptibility maps [6–9]. As increasing attention is paid to the application of remote sensing and geographic information system technology, many scholars have used quantitative and qualitative methods based on expert group experience, to prepare landslide susceptibility maps [10,11]. Generally, a qualitative method uses a landslide inventory to predict sites with the same geomorphological and geological characteristics, or to combine the concept of weighting and sequencing with the view of experts to predict potential landslides [12–14]. A quantitative method is a numerical

analysis of spatial relationships of historical landslides and conditioning factors [13]. The qualitative method based on expert knowledge may be too subjective, so the quantitative approach is increasingly being applied to landslide susceptibility modeling [13,15,16].

There are three kinds of quantitative methods: physically-based methods, statistical-based methods, and machine learning methods. Physically-based methods [17,18] require more detailed data of rock and soil records and regional-scale data relating to the geological aspects of slope failure at specific locations [19,20]. However, because of the associated high cost, this approach is not suitable for large areas. Traditional statistical analysis requires the presumption of a structural model. It then focuses on parameterization, which is widely applied to the study of geological disasters, such as landslides [19,21,22]. However, this method is not very practical, and has a number of defects. In particular, it requires an underlying hypothesis, and has an over-dependence on parameters [19], such as the certainty factor (CF) [23,24], Dempster–Shafer theory [25,26], entropy [23,27], and weight of evidence (WoE) [28–30].

The machine learning approach includes a series of data-driven algorithms and tools that are used to study the relationship between the occurrence of landslides and the conditioning factors related to landslides [31,32]. These methods overcome the shortcomings of physically based and statistical methods, and produce more reliable results from data [33–35]. In general, the machine learning method is more effective than other methods [33,34], which is a common feature of the machine learning algorithm. For instance, support vector machines (SVMs) [36,37], artificial neural networks (ANNs) [38–40], random forest [41–43], naive Bayes (NB) classifiers [35], and other machine learning models are widely used in LSM. Pourghasemi and Rahmati conducted a comparative study of the 10 most advanced machine learning models [19].

It can be clearly seen from the literature review that additional comparative studies of distinct machine learning models are required for better problem analysis, evaluating the effectiveness of models, and to help improve landslide susceptibility prediction [19,34,44]. A large number of studies have been undertaken to explore the landslide susceptibility of the high-risk landslide disaster area of Nanchuan District (the research area). In this paper, the most suitable model and algorithm are identified. The main focus of this paper is to utilize the evidential belief function (EBF), function tree (FT), logistic regression (LR), and logistic model tree (LMT) to perform a landslide susceptibility forecast in Nanchuan District, and subsequently to compare the performances of the four models.

## 2. Description of the Study Area

Nanchuan District is part of Chongqing, China, and is located at 106°54′–107°27′ E and 28°46′–29°30′ N. Nanchuan District is adjacent to Wulong in the northeast, Guizhou in the southeast, Fuling in the north, and Banan, Qijiang and Wansheng in the west. It is the necessary transportation gateway of Southern Chongqing and Northern Guizhou. The district is 80.25 km in length from south to north, 52.5 km wide from east to west, and has a total area of 2601.92 km$^2$ (Figure 1). Furthermore, Nanchuan District is located in a subtropical warm monsoon area with abundant rainfall and a mild climate. The annual average temperature is 20 °C. The maximum annual rainfall is 1534.80 mm, and the annual average rainfall is 1170.20 mm. Figure 2 shows the average monthly rainfall from 1990 to 2019. The seasonal distribution of rainfall is uneven, the driest month is January, and the rainy season is mainly from May to September [45,46] (http://www.weather.com.cn/).

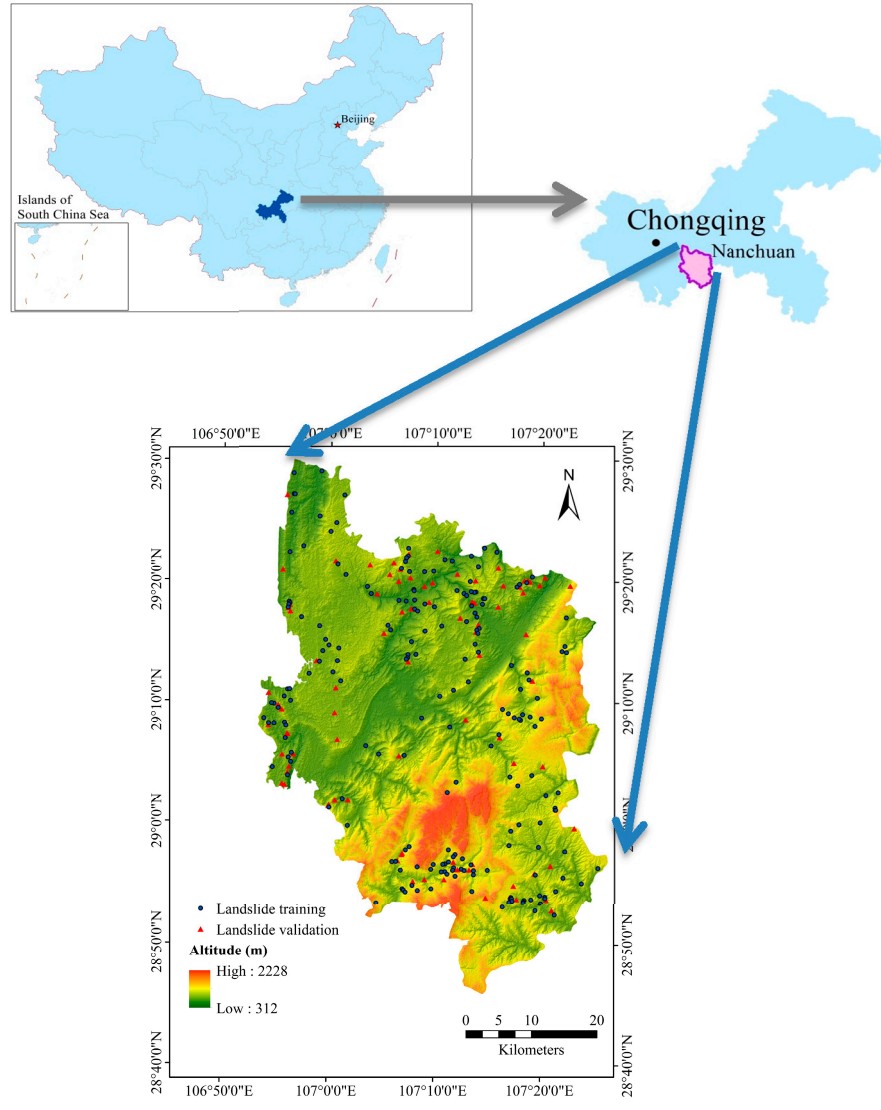

**Figure 1.** The study area.

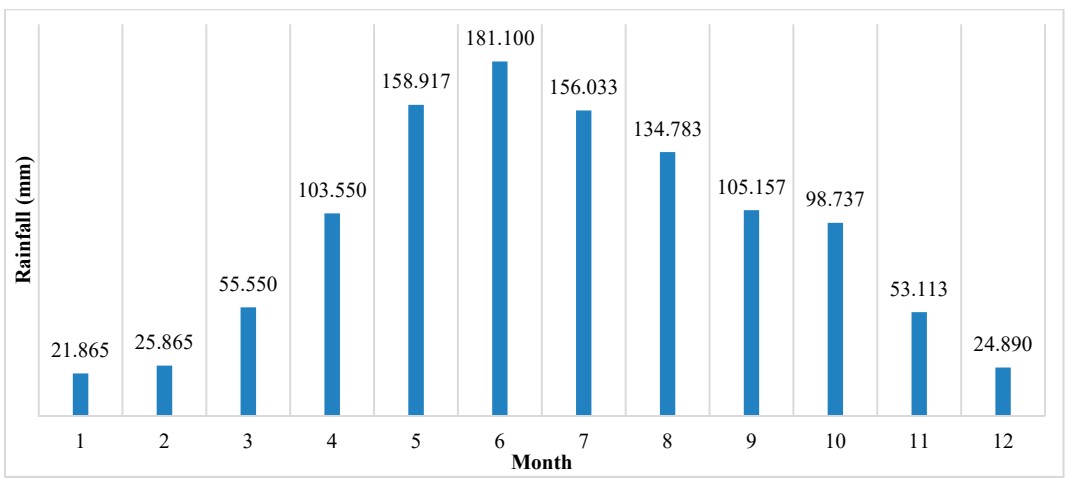

**Figure 2.** Average monthly rainfall from 1990 to 2019.

Nanchuan District is located on the southeastern margin of the Sichuan Basin, and to the northwest of the Dalou Mountains. The area has the characteristics of strong settlement in the northwest (within the basin) and uplift in the southeast. The lithology in the area is mainly mudstone, sandstone, and limestone, and Quaternary deposits are widely distributed in depressions, river valleys, and slopes. Paleozoic and Mesozoic deposits are widely distributed in the area. The lithology is mainly carbonate rocks and clastic rocks, and there are a few Quaternary loose accumulation layers, which laid the foundation for the formation of groundwater and constituted carbon. The three basic types of groundwater are karst water in salt rock, fissure water in bedrock, and pore water in loose rock. The surface water system in the area mainly comprises the Yangtze River system, which is mostly branched, followed by feathers, with a large slope drop and rapid water flow.

## 3. Methodology

This paper is mainly divided into five parts, as shown in Figure 3.

### 3.1. Evidential Belief Function

The EBF is primarily based on the Dempster–Shafer evidence algorithm theory [47,48]. The Dempster–Shafer theory, as an extension of Bayesian subjective probability theory, mainly deals with the influence of the confidence degree of the problem and the probability of the problem.

The main advantage of applying this method to a landslide susceptibility study is its adaptability. This is mainly due to the integration of beliefs from multiple sources and the acceptability of uncertainty. The other advantage of the EBF model is that it is a method of uncertainty prediction in the landslide mapping area [25]. These advantages lead to the EBF model having good prediction results as a two-variable model. Belief (Bel) multi-tier integration is represented by the following formula [49]:

$$Bel = \frac{Bel_1 + Bel_2 + \cdots + Bel_n}{1 - \sum_{i=2}^{n} Bel_{i-1}Dis_i - Dis_{i-1}Bel_i} \tag{1}$$

where $Bel_n$ indicates the element of each type or range of low confidence, $Bel$ denotes the lower limit of the propositional probability, and $Dis_i$ indicates the level of distrust for each factor type or scope. Thus, if there is no landslide, the $Bel$ value will be zero.

### 3.2. Function Tree

As an effective classification method, the FT was first considered as a multivariate tree for the promotion of decision problems [50]. Gama put forward the FT model, which combines a significant construction discriminant function with a multivariate decision tree [50,51]. Here, $D$ is the training dataset and n is the number of samples $(X_i, Y_i)$, $X_i \in \mathbb{R}^n$, $Y_i \in \{1, 0\}$. $Xi$ is an input variable, which, in this paper, refers to the 16 landslide conditioning factors. $Y_i$ is an output, which can be expressed as landslide or non-landslide. FT first establishes a decision tree to separate these two classes from the training dataset [52]. The FT algorithm uses a logistic regression function to segment the internal nodes (called oblique crack) and make an estimate on the leaves. Then, $P(X)$ is the predictive value (PV) of the measured probability and the logical enhancement of the iteratively reweighted least squares method is determined for each class $Y_i$ (for each output comprising two classes) [41,52].

$$f_{Y_i}(X) = \sum_{i=1}^{15} \beta_i X_i + \beta_0 \tag{2}$$

$$P(X) = \frac{e^{2f_{Y_i}(X)}}{1 + e^{2f_{Y_i}(X)}} \tag{3}$$

where $X_i$ is the input variable and $\beta_i$ is the modulus of the *i*-th component.

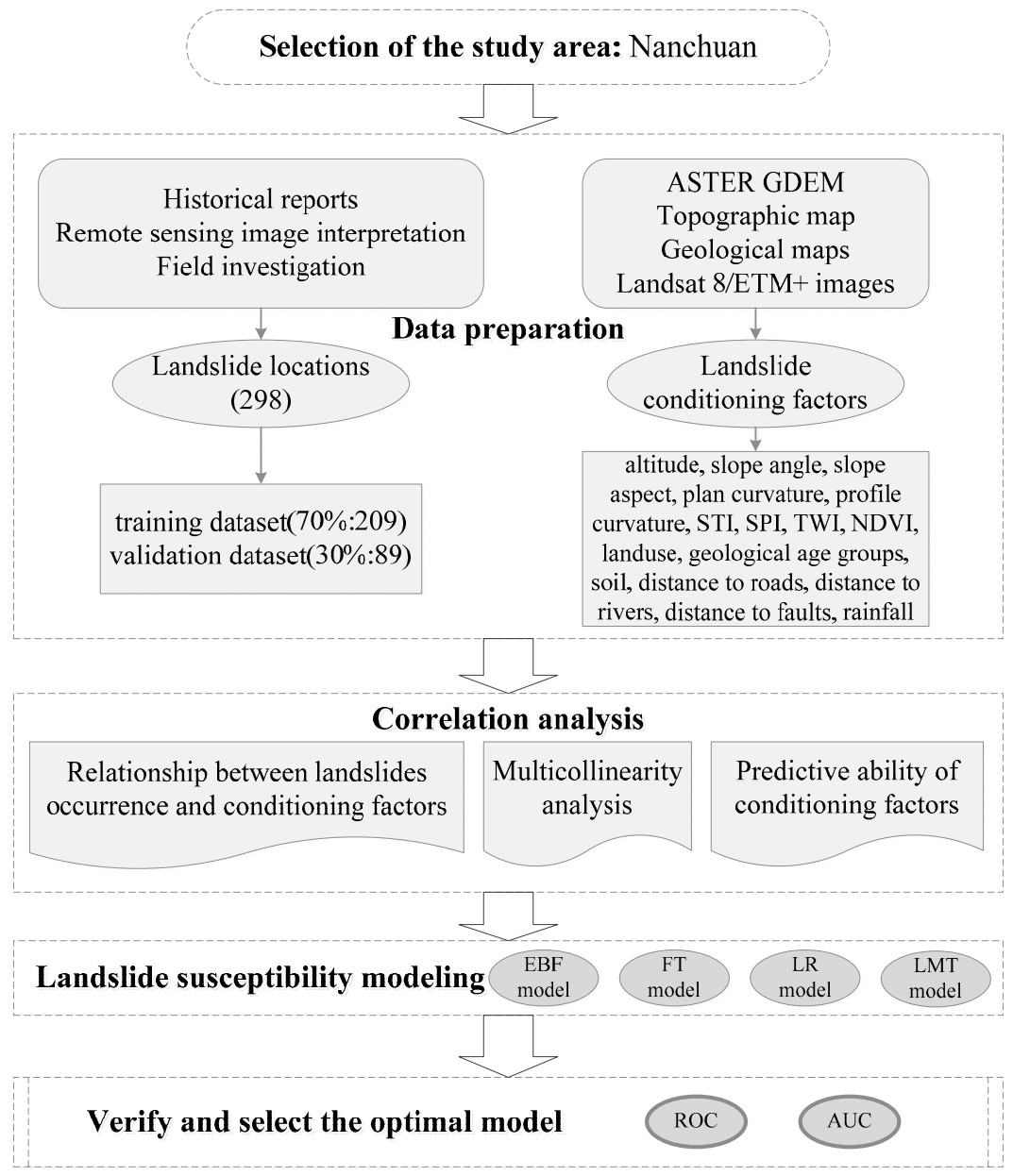

**Figure 3.** The flowchart used in this research.

### 3.3. Logistic Regression

LR relates to one or more independent variables to predict the binary classification or reaction probability [29,53]. Values of arguments in the model can be represented by 0 or 1, indicating whether landslide units exist or not. The general form of LR is as follows [54]:

$$y = a + b_1x_1 + b_2x_2 + b_3x_3 + \cdots + b_mx_m \tag{4}$$

$$y = log_e[\frac{P}{1-P}] = logit(P) \tag{5}$$

$$P = \frac{e^y}{1 + e^y} \tag{6}$$

where $x_1, x_2, \ldots, x_m$, from $X_i$, represent arguments; $b_1, b_2, \ldots, b_m$ represents the estimated regression coefficients; $y$ denotes a combination function with a linear relationship; and $P$ represents the probability of landslide.

### 3.4. Logistic Model Tree

As a categorical model, LMT consists of a decision tree learning model and LR [55,56]. The LR of tree nodes is constructed using the LogitBoost algorithm and pruned by the CART algorithm [57,58]. The LogitBoost algorithm selects the most relevant attributes or variables in the data, uses a simple regression in each iteration, and stops working before convergence to the maximum likelihood solution occurs [59]. The LR model is constructed by a staged fitting process, and the relevant attributes in the data are selected. Furthermore, the LogitBoost algorithm creates the accumulated LR of the least-squares method for the preset data of every c class, in the following format [59]:

$$L_c(x) \ = \ \alpha_0 + \sum_{i\,=\,1}^{F} \alpha_i x_i \tag{7}$$

where $\alpha_i$ is the ratio of the *i*-th weight in the observation $x$, and $F$ is the landslide specific number. The probability of nodes in the LMT model can be calculated by the linear LR function [59]:

$$p(c|x) \ = \ \frac{\exp(L_c(x))}{\sum_{c'\,=\,1}^{c} \exp(L_{c'}(x))} \tag{8}$$

where $c$ is the density of landslide classification; $L_c(x)$ is converted to $\sum_{c\,=\,1}^{c} L_c(x) \ = \ 0$ to apply the least squares method [59].

## 4. Data Preparation

### 4.1. Landslide Inventory

The quality of the landslide inventory determines the results of landslide susceptibility prediction and evaluation. However, there is currently no detailed standard for the accuracy of landslide inventory [46,60]. This study uses landslide events caused mainly by multiple rainfalls during the time span of 1979 to 2018. A total of 298 landslides were identified, based on field investigations, historical reports, and Google Earth satellite image interpretations. A complete landslide inventory map of Nanchuan District was used to identify and record the location (centroid) of these previous landslides (Figure 1), which consists of 295 slides and three rockfalls [61]. According to the analysis of landslides using GIS tools, the volumes of the three rockfalls are 4800 m$^3$, 12,000 m$^3$ and 13,100 m$^3$, respectively. The size distribution of 295 slides is shown in Figure 4. For slides, the smallest area was close to 70 m$^2$, the largest was more than $8.4 \times 10^5$ m$^2$, and the average area was about $3.07 \times 10^4$ m$^2$. In terms of volume, the minimum volume was only 140 m$^3$, and more than 95% of the slides were less than 100,000 m$^3$. The occurrence of landslides is closely related to the exposed strata, and its lithological conditions are the decisive factors that determine landslides' occurrence. Landslides are prone to occur in strata distribution areas, such as clay, mudstone, shale, and marl. Based on the above, it can be shown that the established landslide inventory map is sufficiently robust and can be used for the landslide susceptibility analysis in this study. The split of the dataset is of significance to verify the performance of the model [62]. Furthermore, through the analysis and comparison of landslide data, it was found that 70%: 30% could be used as the classification standard of landslide data in this paper [63–65]. In addition, an equal amount (298) of non-landslide points were randomly selected in areas without landslides, and then allocated 70%: 30% to the training and verification data sets, respectively. Then, data were assigned values of 1 (with landslide) and 0 (non-landslide).

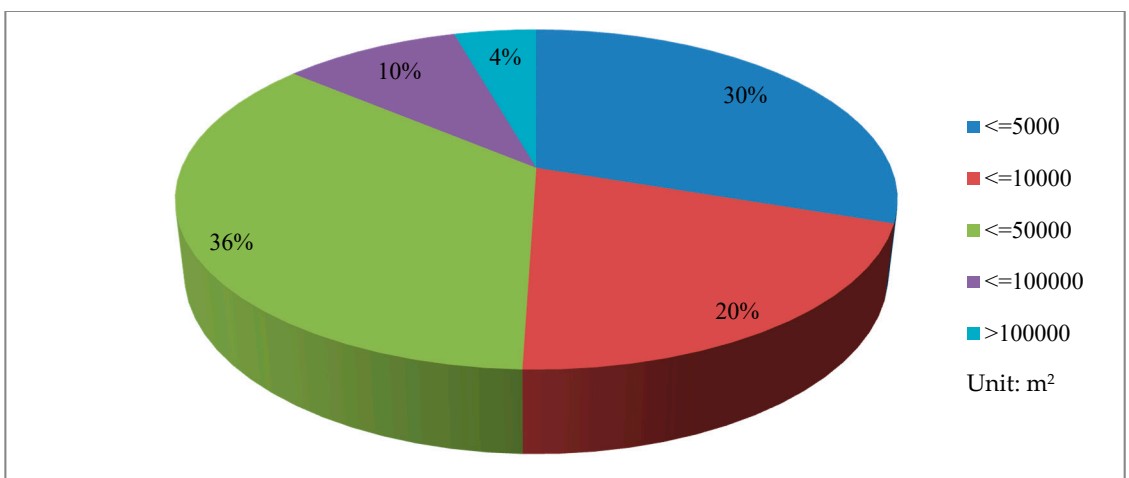

**Figure 4.** The frequency distribution of 295 slides.

## 4.2. Landslide Conditioning Factors

After the compilation of the landslide inventory map, it is necessary to select and create landslide conditioning factors for landslides susceptibility prediction [28]. These factors are mainly selected according to three aspects: geological factors, topographic factors, and geological environment factors. Based on the existing characteristics and geological conditions of the research area and the literature review [66–69], 16 conditioning factors were selected for this paper: altitude, slope angle, slope aspect, plan curvature, profile curvature, sediment transport index (STI), stream power index (SPI), topographic wetness index (TWI), the normalized difference vegetation index (NDVI), land use, geological age groups, soil, distance to roads, distance to rivers, distance to faults, and rainfall (Figure 5). These 16 conditioning factors were converting into a thematic data layer with a unified resolution of 20 × 20 m, in order to achieve the purpose of a unified format, which is conducive to the prediction of landslide susceptibility.

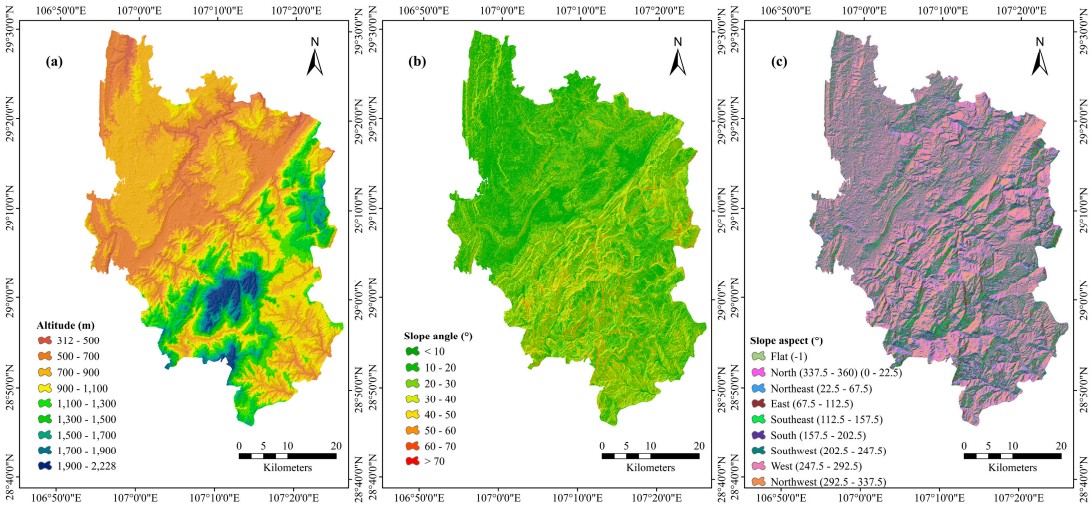

**Figure 5.** *Cont.*

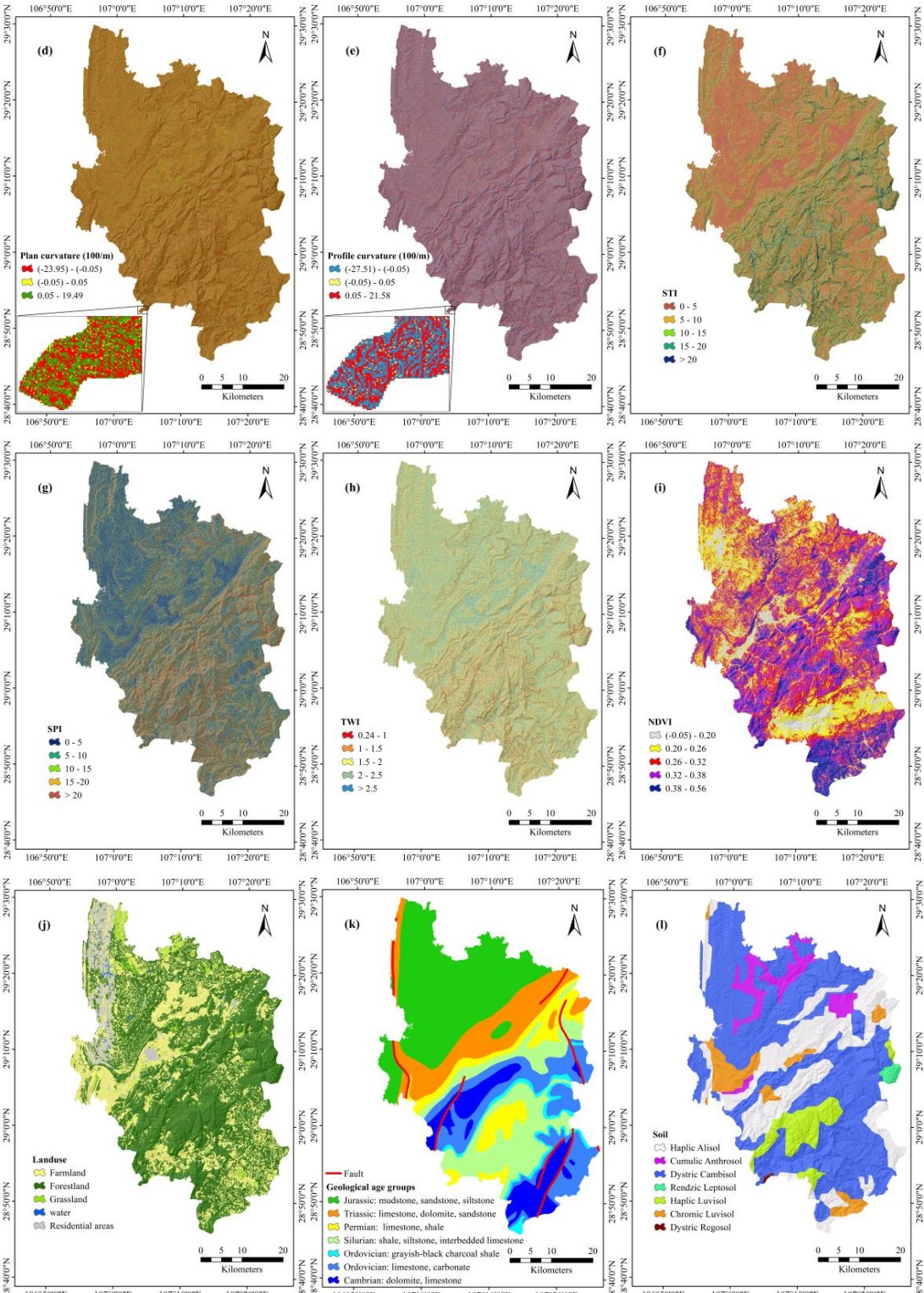

**Figure 5.** *Cont.*

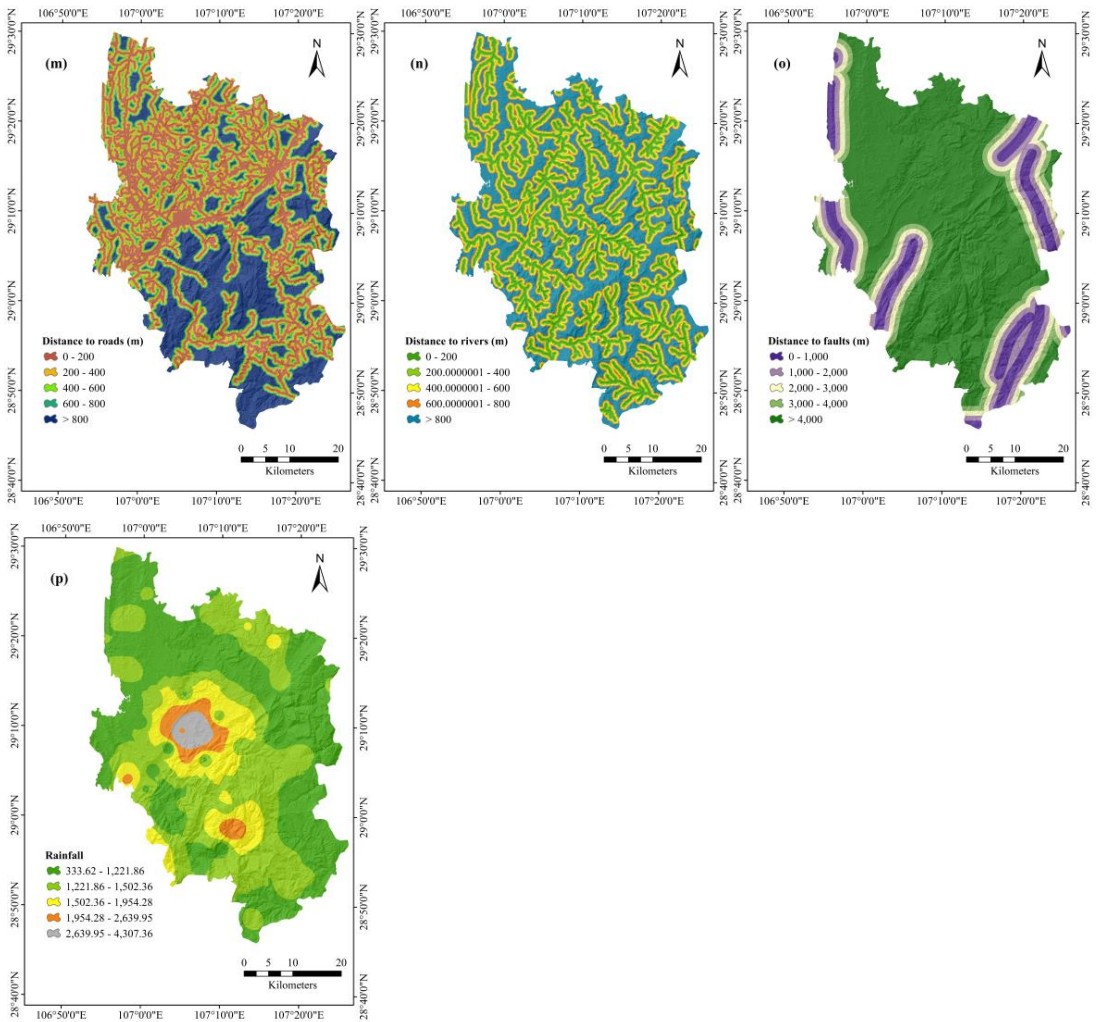

**Figure 5.** Landslide conditioning factors: (**a**) altitude, (**b**) slope angle, (**c**) slope aspect, (**d**) plan curvature, (**e**) profile curvature, (**f**) sediment transport index (STI), (**g**) stream power index (SPI), (**h**) topographic wetness index (TWI), (**i**) normalized difference vegetation index (NDVI), (**j**) land use, (**k**) geological age groups, (**l**) soil, (**m**) distance to roads, (**n**) distance to rivers, (**o**) distance to faults, (**p**) rainfall.

A digital elevation model (DEM) was extracted from Aster GDEM data (http://www.gscloud.cn). In ArcGIS 10.5, the DEM was utilized to extract thematic data layers such as elevation, slope angle, aspect, plan curvature, profile curvature, STI, SPI, and TWI. Landsat 8 OLI images, traffic maps, and 1:200,000 geological maps were used to extract the NDVI, distance to roads, distance to rivers, and distance to faults. Using the rainfall data of Chongqing Meteorological Bureau (http://www.weather.org.cn), the rainfall map was drawn based on the distance weighted inverse method [64,65]. The land-use thematic data were drawn from the 1:100,000 land-use map. In addition, the soil thematic data were extracted using the 1:1,000,000 scale soil map.

## 5. Results

### 5.1. Analysis of Landslide Conditioning Factors

#### 5.1.1. Relationship between Landslide Conditioning Factors and Landslide Occurrence

In this paper, the thematic layer of landslide data and the thematic map of 16 conditioning factors were combined to compute the number of pixels and landslides under different categories, and the proportion of each category was calculated. On this basis, the EBF model can be used to summarize the spatial correlation

between landslide occurrence and landslide conditioning factors (Figure 6). When the *Bel* value is zero, the expected chance of landslide is also zero. Hence, the *Bel* value is positively correlated with the expected probability of landslide occurrence.

The conditioning factor of altitude (Figure 6a), which is divided into nine categories, has the largest *Bel* value (0.235) for 900–1100 m, with no landslides occurring at altitudes higher than 1700 m, and up to the highest elevation in the research area. At the same time, landslides mainly occur at an altitude of 1300 m, which is consistent with the opinion of Wu et al. [70]. In the eight categories classified according to the slope angle (Figure 6b), although the *Bel* value is the largest in the range of 60–70° (0.330), this range only accounts for 0.17% of the total area, and landslides in this range only account for 0.47% of the total. The *Bel* value is only 0.164 in the range of 10–20°, accounting for 35.95% of the total area. Regarding geology, regions in the range of 10–20° are most prone to landslides, due to their instability after heavy rain [71]. Regarding the slope aspect (Figure 6c), the *Bel* value of the southwest (0.148) is the largest, and that of the flat is zero. The slope direction is positively correlated with the occurrence of a landslide, which is consistent with the viewpoint of Oh et al. [72]. For the three types of plan curvature (Figure 6d), the maximum *Bel* value (0.401) occurs in the range of −0.05 to 0.05, while the region of 0.05–19.49 has the smallest *Bel* value (0.279). The area of the region −0.05 to 0.05 is small, so the confidence is not large, which may also be related to the overweight effect [73,74]. Among the three types of profile curvature (Figure 6e), −27.51 to −0.05 has the highest *Bel* value (0.359), and −0.05 to 0.05 has the lowest *Bel* value (0.301). However, for the profile curvature range of −27.51 to −0.05, the susceptibility is higher, which is consistent with Hong et al. [75]. Furthermore, the *Bel* value of the STI is the largest for STI >20 (0.318) of the seven categories and the smallest for 15–20 (0.087) (Figure 6f). The SPI is divided into five categories, and the largest *Bel* value was obtained for SPI >20 (0.256) (Figure 6g). The STI and SPI are positively correlated with the rate of occurrence of landslides (with the exceptions of 15–20 and 5–10, respectively), as in [76]. For the TWI (Figure 6h), 0.24–1 (0.225) and 1–1.5 (0.178) had the largest and smallest *Bel* values, respectively, of the five constituent categories. The lowest *Bel* value of the NDVI appeared for 0.38–0.56 (0.140), while the highest *Bel* value appeared for 0.20–0.26 (0.231) (Figure 6i). In the five categories of land use (Figure 6j), farmland has the largest *Bel* value (0.424). As for land use, the susceptibility of farmland is the largest, which is closely related to land irrigation, human engineering activities, and rainfall. Furthermore, for geological age groups (Figure 6k), the area of the fifth of the seven groups (Ordovician: greyish-black charcoal shale, siliceous base) is prone to landslides with the largest *Bel* value (0.650)). This is consistent with the opinion of Jaafari et al. [77] that shale transports permeated water to the fracture surface. Soil is divided into seven classes (Figure 6l), with the largest *Bel* value obtained by the Dystric Cambisol class (0.363), while the *Bel* values of the Rendzic Leptosol and the Dystric Regosol classes were zero. Regarding the distance to roads (Figure 6m), the 0–200 m group has the maximum *Bel* value (0.296) of the five categories. Here, the underlying trend is that the higher the distance to roads, the less prone an area is to landslides. Among the five categories of distance to rivers, the *Bel* value of the 200–400 m group was the highest (0.260) (Figure 6n). Skilodimou [78] proposed that urban and rural planners and engineers can use a certain distance of a buffer zone to reduce the occurrence of landslides and protect the existing environment. For the distance to faults (Figure 6o), the maximum (0.247) and minimum (0.144) *Bel* values were obtained for the 1000–2000 m and 2000–3000 m groups of the seven classes, respectively. In terms of rainfall (Figure 6p), the largest *Bel* value (0.340) was obtained in the 333.62–1221.86 class.

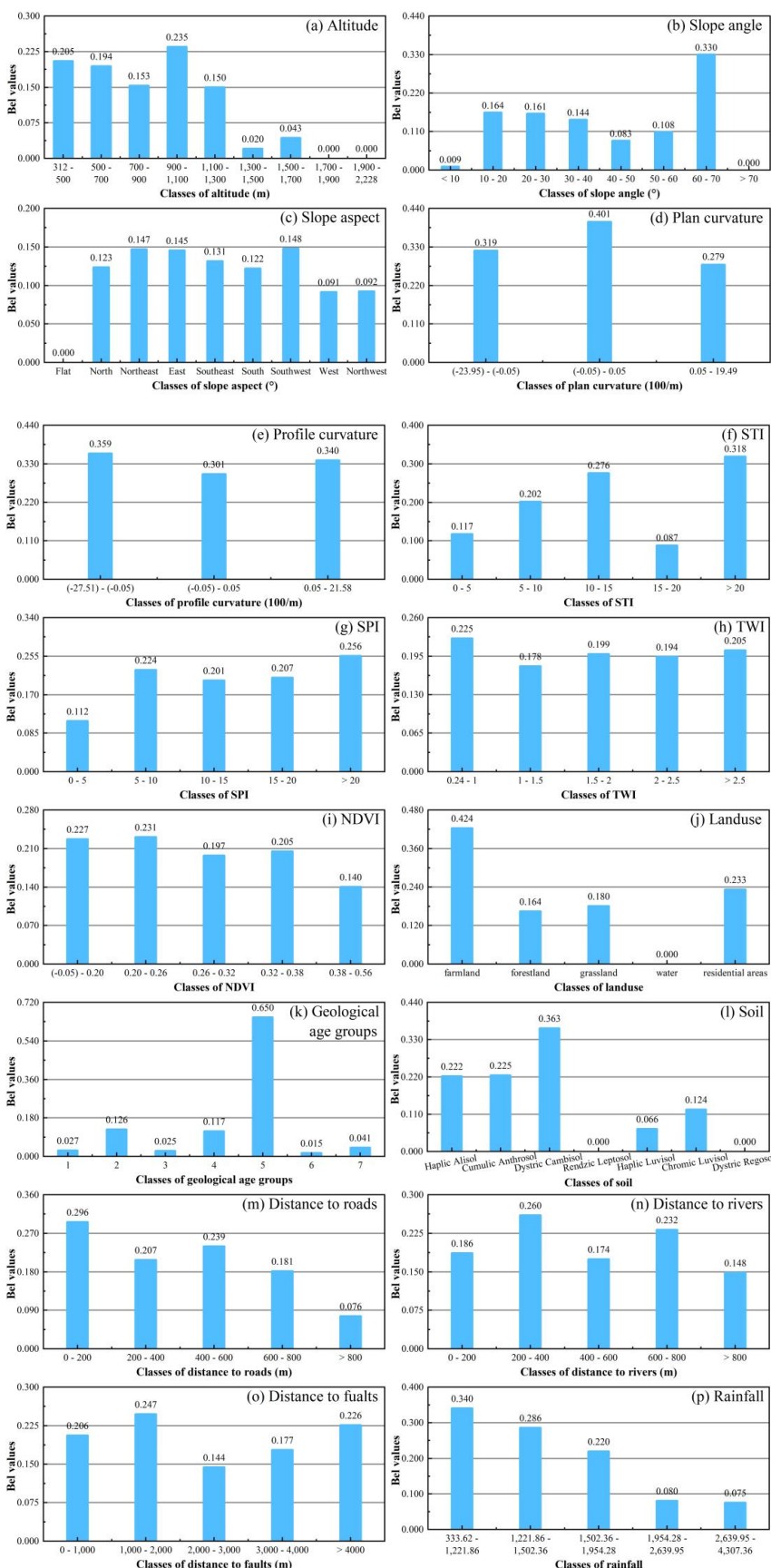

**Figure 6.** Relationship between landslides and the landslide conditioning factors of (**a**) altitude, (**b**) slope angle, (**c**) slope aspect, (**d**) plan curvature, (**e**) pProfile curvature, (**f**) STI, (**g**) SPI, (**h**) TWI, (**i**) NDVI, (**j**) land use, (**k**) geological age groups, (**l**) soil, (**m**) distance to roads, (**n**) distance to rivers, (**o**) distance to faults, (**p**) rainfall.

### 5.1.2. Multicollinearity Analysis of Conditioning Factors

In the process of building a landslide susceptibility model, it is necessary to carry out multicollinearity analysis to test the conditioning factors [79,80]. Multicollinearity can be used to test the possible interdependency between the conditioning factors of a landslide. If there is a high degree of correlation, it will lead to serious system analysis error. Generally speaking, the tolerance and variance inflation factor (VIF) are the commonly used indicators of the multicollinearity test (tolerance < 0.1 or VIF > 10) [81,82]. The expressions for tolerance and VIF are as follows:

$$Tolerance = 1 - R^2{}_d \tag{9}$$

$$VIF = \left[ \frac{1}{Tolerance} \right] \tag{10}$$

where $R^2{}_d$ is the determining factor for the regression of explanatory variables and $d$ concerns all other explanatory variables [67,83].

Therefore, a multicollinearity analysis was carried out to determine if there is interdependence between the adjustment factors of the EBF model preprocessing. The two values of tolerance and VIF were obtained by multicollinearity regression modeling (Table 1). From Table 1, the lowest VIF value (1.052) and the highest tolerance value (0.951) are exhibited by the distance to the rivers conditioning factor. In contrast, the SPI conditioning factor has the largest VIF value (2.157) and the smallest tolerance value (0.464).

**Table 1.** Multicollinearity analysis in the current study.

| Landslide Conditioning Factor | Collinearity Statistics | |
|:---:|:---:|:---:|
| | **Tolerance** | **VIF** |
| Distance to rivers | 0.951 | 1.052 |
| Slope aspect | 0.936 | 1.069 |
| Distance to faults | 0.907 | 1.103 |
| Rainfall | 0.903 | 1.108 |
| Profile curvature | 0.896 | 1.116 |
| Land use | 0.886 | 1.129 |
| Plan curvature | 0.871 | 1.148 |
| Geological age groups | 0.863 | 1.158 |
| Distance to roads | 0.855 | 1.169 |
| NDVI | 0.845 | 1.183 |
| Slope angle | 0.84 | 1.19 |
| Soil | 0.804 | 1.244 |
| TWI | 0.769 | 1.300 |
| Altitude | 0.767 | 1.304 |
| STI | 0.536 | 1.867 |
| SPI | 0.464 | 2.157 |

The results show that all of these factors meet the conditions of TOL value greater than 0.1 and VIF value less than 10 [29,79,84]. Therefore, no multicollinearity issues exist between the 16 conditioning factors in the current paper.

### 5.1.3. The Prediction Ability of the Conditioning Factors

The removal of negative landslide susceptibility factors can optimize model performance and ensure the accuracy of landslide susceptibility prediction [85]. In the current paper, the prediction ability of the conditioning factors pretreated by the EBF model was investigated and quantified by the contribution values of the 16 landslide conditioning factors. Ten-fold cross-validation and the correlation attribute evaluation method (CAE) were used to calculate the average merge and standard deviation of each conditioning factor [86–88]. The results show that the slope angle

(AM = 16) has the greatest influence on landslide occurrence, followed by the altitude (AM = 15), distance to roads (AM = 13.4), soil (AM = 12.6), rainfall (AM = 12.4), geological age groups (AM = 10.8), STI (AM = 10.6), distance to faults (AM = 8.6), land use (AM = 8.6), distance to rivers (AM = 6.7), slope aspect (AM = 5.9), SPI (AM = 4.3), NDVI (AM = 3.3), TWI (AM = 2.9), profile curvature (AM = 2.8), and plan curvature (AM = 2.1) (Figure 7). Each conditioning factor has a corresponding predictive value. Hence, all of these factors were adopted in this study.

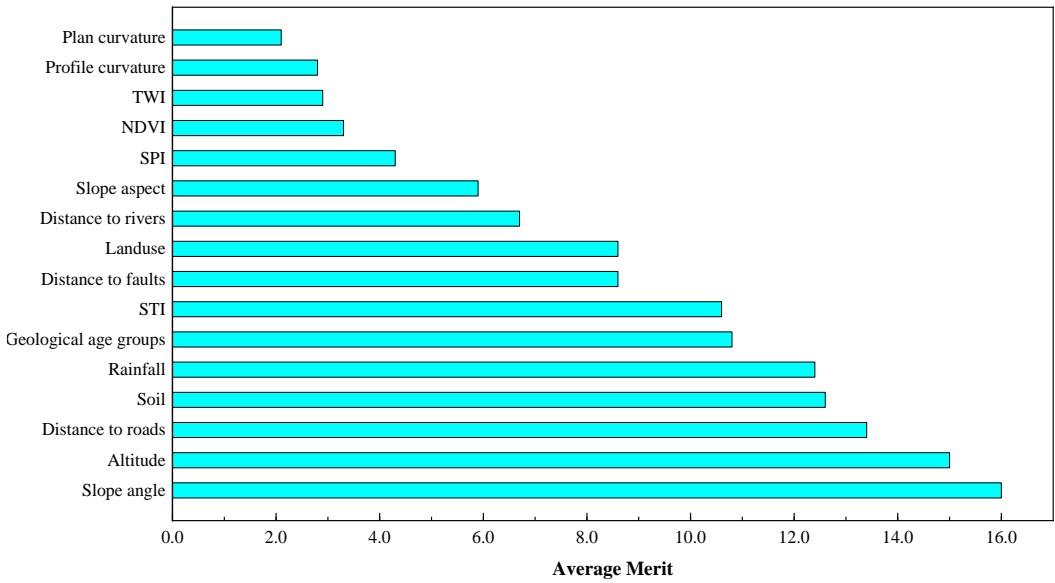

**Figure 7.** The prediction ability of the landslide conditioning factors.

### 5.2. Model Configuration

In the current paper, the landslide susceptibility models were analyzed by EBF, FT, LR, and LMT models. The EBF algorithm was mainly used to preprocess the landslide data in this paper. The landslide susceptibility index (LSI) of the EBF model is calculated, as in Equation (11):

$$
\begin{aligned}
LSI_{EBF} = \quad & Altitude_{Bel} + Slope\ angle_{Bel} + Slope\ aspect_{Bel} + \\
& Plan\ curvature_{Bel} + Profile\ curvature_{Bel} + STI_{Bel} + \\
& SPI_{Bel} + TWI_{Bel} + NDVI_{Bel} + Landuse_{Bel} + Geological\ age\ groups_{Bel} + \\
& Soil_{Bel} + Distance\ to\ roads_{Bel} + Distance\ to\ rivers_{Bel} + \\
& Distance\ to\ faults_{Bel} + Rainfall_{Bel}
\end{aligned}
\tag{11}
$$

The FT model was used as one approach to perform the landslide susceptibility assessment. After the preprocessing of the conditioning factors, the FT algorithm was implemented using the Weka software. The FT model can be applied in three ways: including leaves and inner nodes, only inner nodes, or only leaves. In this paper, after three ways of testing, it was found that only using the leaves yielded the best results.

Thus, it was decided to implement the FT model using only the leaves in this work. As the output representation style of the FT model, the classification decision tree highly generalizes, organizes, and analyzes the landslide data and the conditioning factors under the FT model, classifies the landslide data through the LR model, and gradually forms the binary tree structure (Figure 8). The classification decision tree of the FT model shows that the final output contribution (weight) corresponds to the initial output contribution, which is consistent with Freund's and Mason's views [89].

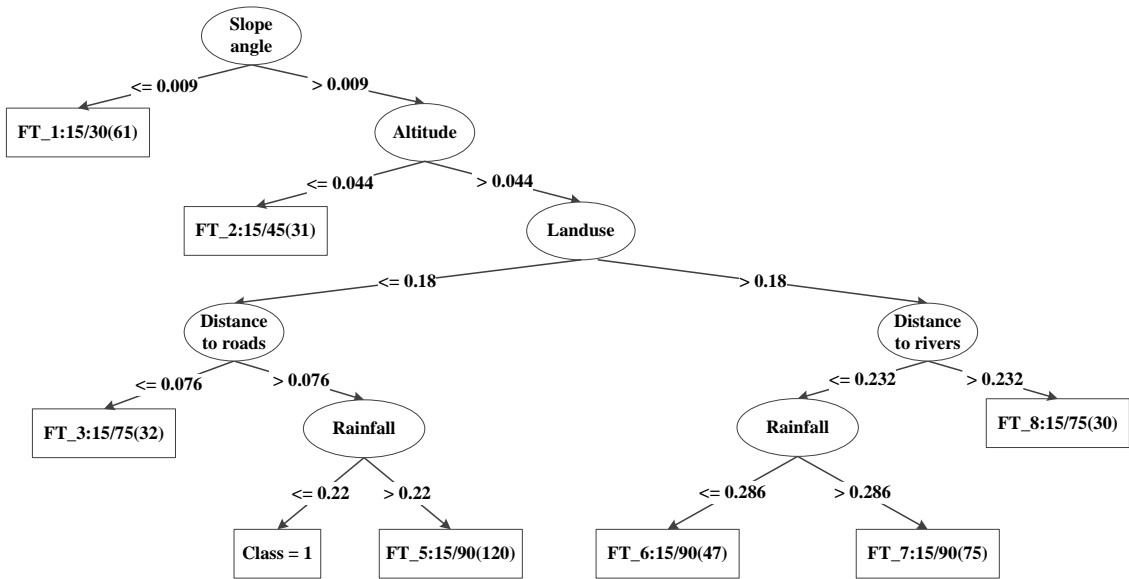

**Figure 8.** Classification decision tree of landslide susceptibility using the function tree (FT) model.

In landslide susceptibility mapping, the LR model quantizes the contribution of the 16 conditioning factors into coefficients to quantify the LSI. The expected probability of landslide occurrence depends on the value of the coefficient. The larger the coefficient value of the conditioning factor, the higher the influence of the conditioning factor on landslide susceptibility. A negative value of the coefficient indicates that the factor reduces the probability of landslide occurrence. There may also be intercepts that affect LSI values. The LSI value range is [0,1]; a value approaching 1 indicates a landslide, while that approaching 0 indicates no-landslide.

Based on the dependent variables (landslide data) and independent variables (16 conditioning factors), the values of various conditioning factors in the LR model are calculated using the Weka software. The selected test mode is 10-fold cross-validation. Meanwhile, the linear combination equation of the LSI of the LR model was constructed (Equation (12)). The intercept (−19.0897) and the coefficient of every conditioning factor is shown in the formula. The coefficient size of each factor is different, and the contribution to LSI is also different. From these coefficients, in the LR model, slope angle and TWI have the greatest impact on LSI, while the negative NDVI and profile curvature have a suppressive effect on LSI.

$$
\begin{aligned}
LSI_{LR} = \ & (Altitude_{Bel} * 9.633) + (Slope\ angle_{Bel} * 18.722) + \\
& (lope\ aspect_{Bel} * 14.034) + (Plan\ curvature_{Bel} * (-0.717)) + \\
& (Profile\ curvature_{Bel} * (-3.059)) + (STI_{Bel} * 5.289) + \\
& (SPI_{Bel} * 0.1794) + (TWI_{Bel} * 18.062) + (NDVI_{Bel} * (-4.0862)) + \\
& (Landuse_{Bel} * 4.381) + (Geological\ age\ groups_{Bel} * 7.737) + \\
& (Soil_{Bel} * 2.372) + (Distance\ to\ roads_{Bel} * 7.848) + \\
& (Distance\ to\ rivers_{Bel} * 7.039) + (Distance\ to\ faults_{Bel} * 11.383) + \\
& (Rainfall_{Bel} * 5.876) - 19.0897
\end{aligned}
\tag{12}
$$

To improve the performance of the LMT model, the calculation parameters need to be adjusted. In this study, according to the data obtained in the modeling process and many attempts, three parameters, given in Table 2, were selected for the LMT model by 10-fold cross-validation, and then optimized. Curves of the values of the AUC under different conditions were drawn (Figure 9). Here, two graphs and eight lines are included. The AUC values were observed and calculated. According to the AUC value, the parameters with better performance were selected and applied to

LMT modeling. The AUC value obtained from the best training data parameter was 0.847, and from the best parameter for the validation data was 0.765 (Table 2).

**Table 2.** Relevant parameters of the LMT model.

|  | numBoostingIterations | splitOnResiduals | useAIC |
|---|---|---|---|
| line1 | [−1, 30] | FALSE | FALSE |
| line2 | [−1, 30] | FALSE | TRUE |
| line3 | [−1, 30] | TRUE | FALSE |
| line4 | [−1, 30] | TRUE | TRUE |
| Selected parameters | 0 | TRUE | TRUE |

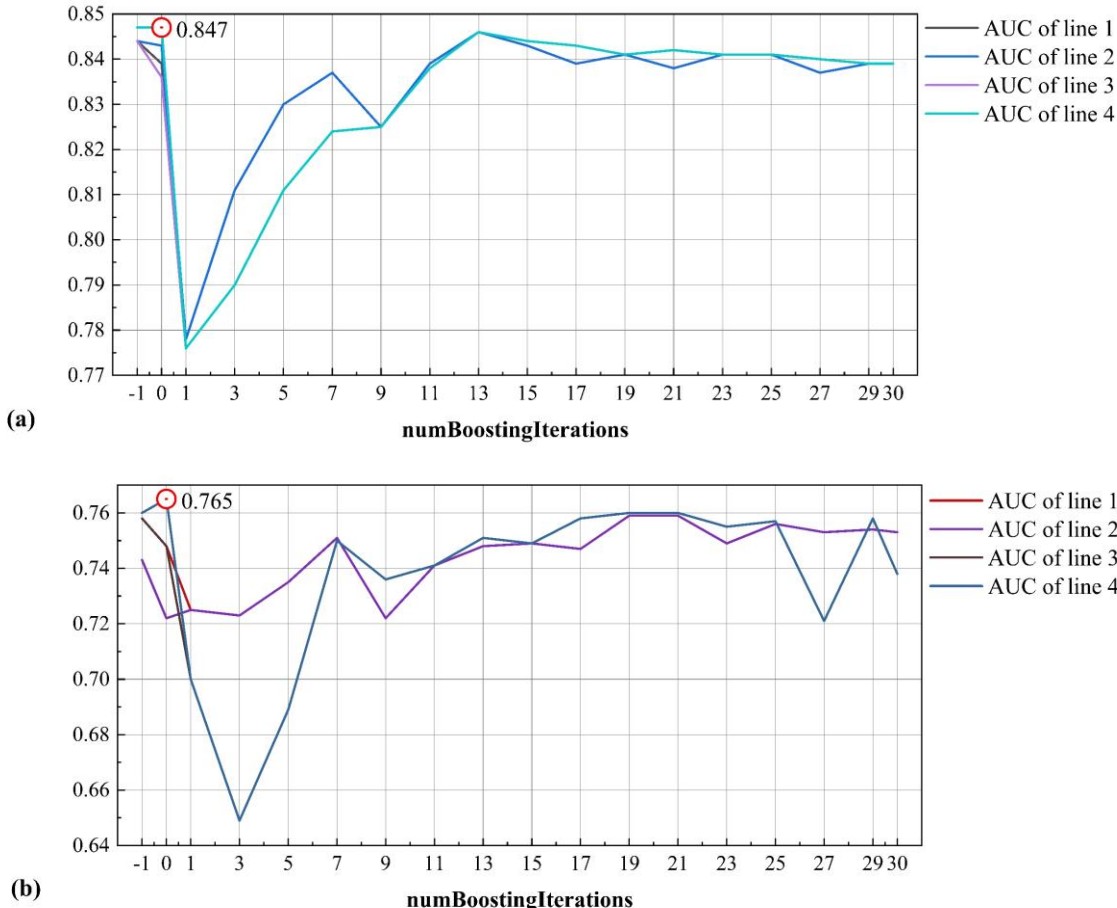

**Figure 9.** Optimization of parameters for the logistic model tree (LMT) model: (**a**) training dataset; (**b**) validation dataset.

## 5.3. Model Validation

The landslide data was assigned a value of 1. In the study area, the pixels equal to the landslide data were randomly sampled as non-landslide data, with a value of 0. The 16 conditioning factors were used to sample these pixels and generate the training and validation datasets. The landslide data preprocessed by the EBF model were applied to the three models (FT, LR, and LMT). Then, the training and validation data sets of these models were calculated in the form of line graphs, and the error statistics were reported (Figure 10). The datasets of the four models for landslide training were statistically analyzed (Figure 11). The mean standard errors (Mean std. error) for the EBF, FT, LR, and LMT models were 0.0066, 0.0193, 0.0156, and 0.0147, respectively. The standard deviations (Std. deviation) of the EBF, FT, LR, and LMT models were 0.135, 0.395, 0.319, and 0.301, respectively.

Meanwhile, the variances of the EBF, FT, LR, and LMT models were 0.018, 0.156, 0.102, and 0.091, respectively. According to the analysis shown in Figures 10 and 11, the best performance was achieved with the EBF model, while the LMT model was better than the LR model and the FT model.

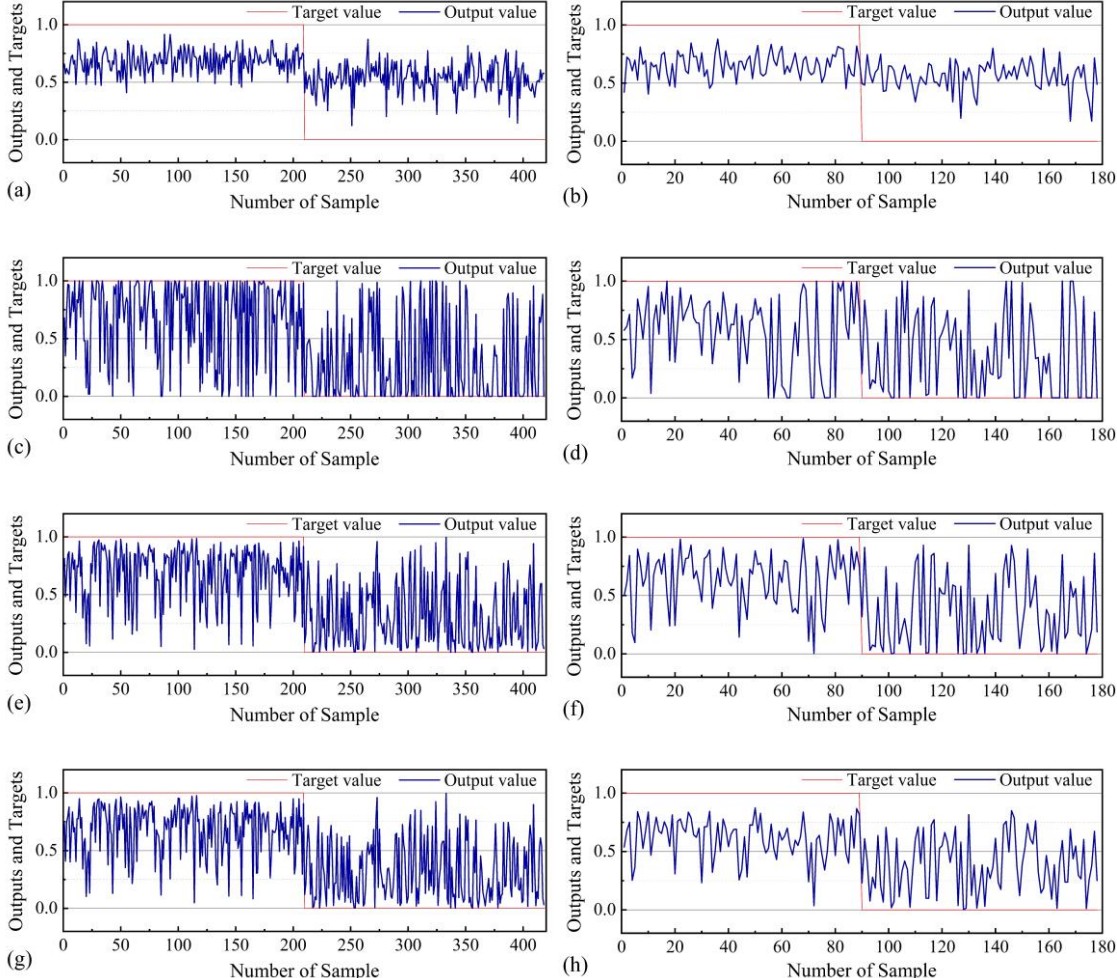

**Figure 10.** Target and output values for the evidential belief function (EBF) model of (**a**) training and (**b**) validation dataset samples; FT model of (**c**) training and (**d**) validation dataset samples; logistic regression (LR) model of (**e**) training and (**f**) validation dataset samples; LMT model of (**g**) training and (**h**) validation dataset samples.

In the current study, ROC curves and AUC values were used to compare and evaluate training and validation datasets of the landslide susceptibility model (Figure 12). The ROC curve is a coordinate graph and a high-quality tool for probability prediction systems [90,91]. In the coordinate system, the closer the point of the ROC curve to the upper left corner, the higher the accuracy of the test results. The AUC value range is [0.5, 1.0], in which the highest AUC has the best diagnostic value [92]. Then, the AUC value produced by an excellent model is between 0.9-1, good model (0.8-0.9), fair model (0.7–0.8), a poor model in the range of 0.6-0.7 and the final 0.5-0.6 poor accuracy range of the model [93]. The AUC of the LMT model was the highest (0.847) in the training dataset, followed by the LR (0.838), EBF (0.824), and FT (0.780) models. In the validation dataset, the highest AUC value was given by the LMT model (0.765), followed by the LR (0.756), EBF (0.737), and FT (0.676) models. These results show that all of the AUC values of the training dataset were slightly higher than those of the validation dataset, and the LMT model was the best of the four models.

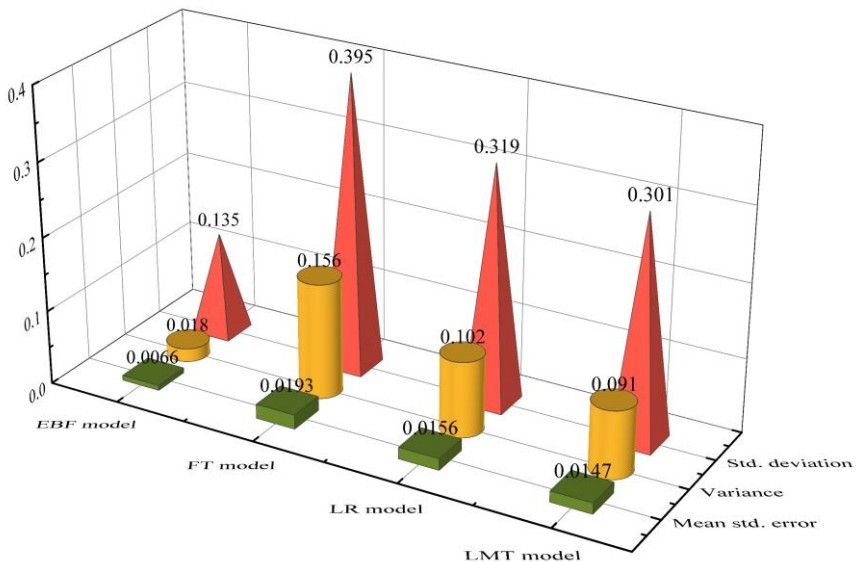

**Figure 11.** Statistical analysis of the performance of the EBF, FT, LR, and LMT models.

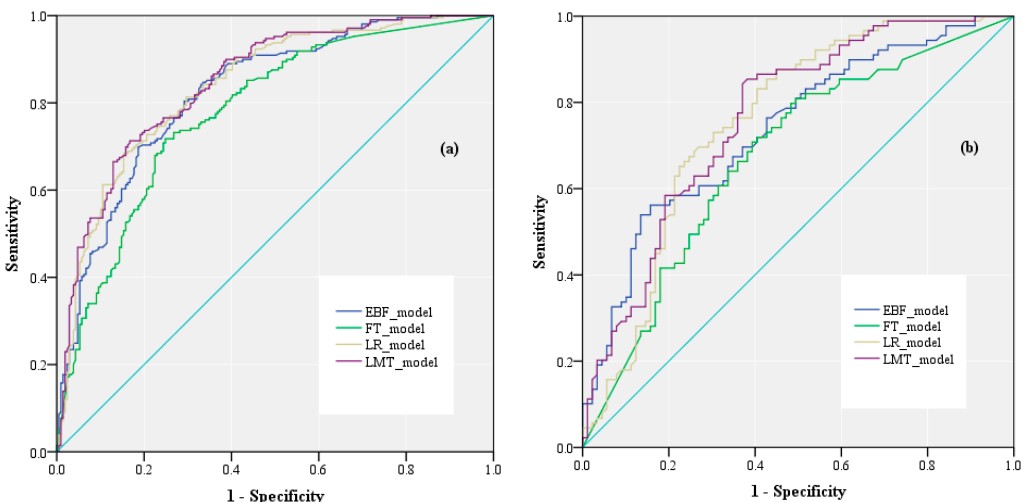

**Figure 12.** Receiver operating characteristic (ROC) curves: (**a**) training dataset; (**b**) validation dataset.

## 5.4. Generating Landslide Susceptibility Maps

In this study, a unique LSI was applied to all pixels in this research area, to establish a landslide susceptibility model. The classification methods of LSI mainly include Jenks natural breaks, quantile, geometrical interval, equal interval, and standard deviation [94,95]. In this study, the four classification methods of Jenks natural breaks, equal interval, quantile, and geometrical interval were used to divide LSI into five categories: very low (VLC), low (LC), moderate (MC), high (HC), and very high (VHC). The relative distribution of susceptibility category areas and the relative proportion of landslides in each category in the study are shown in Figure 13. Generally, the histograms of different models under different classification methods show regularity: the higher the susceptibility category, the greater the landslide distribution. Most landslides were recorded in the very high (VHC) category. It can be clearly seen from Figure 13 that this rule exists, and the quantile is superior to the other three classification methods, with outstanding performance. In the FT model, the quantile classification method does not perform as well as the other three models, but for the FT model, the quantile classification method is still its best choice. Therefore, the quantile is used as a classification scheme for landslide susceptibility maps of the EBF, FT, LR, and LMT models.

The final four landslide susceptibility maps were drawn according to the selected classification method (Figure 14). In the EBF model (Figure 14a), the scales of the VLC (19.75%), LC (19.51%), MC (20.65%), HC (20.51%), and VHC (19.59%) classifications are distributed. The proportions of landslides in the areas of various categories are VLC (1.35%), LC (7.07%), MC (13.80%), HC (22.22%), and VHC (55.56%). In the modeling results of the FT model (Figure 14b), the proportions of VLC (26.52%), LC (23.14%), MC (16.94%), HC (16.58%), and VHC (16.82%) categories are distributed. The proportions of landslides in the areas of each category are VLC (8.08%), LC (7.74%), MC (22.22%), HC (27.61%), and VHC (34.34%). According to the landslide susceptibility zoning results of the LR model (Figure 14c), the scales of the VLC (19.86%), LC (20.47%), MC (19.77%), HC (20.20%), and VHC (19.70%) classifications are distributed. The proportions of landslides in the areas of each category are VLC (1.35%), LC (11.78%), MC (17.17%), HC (23.23%), and VHC (46.46%). Finally, in the modeling results of the LMT model (Figure 14d), the proportions of VLC (20.00%), LC (20.11%), MC (20.24%), HC (19.69%), and VHC (19.96%) categories are distributed. The proportions of landslides in the areas of each category are VLC (1.68%), LC (5.05%), MC (19.53%), HC (28.28%), and VHC (45.45%).

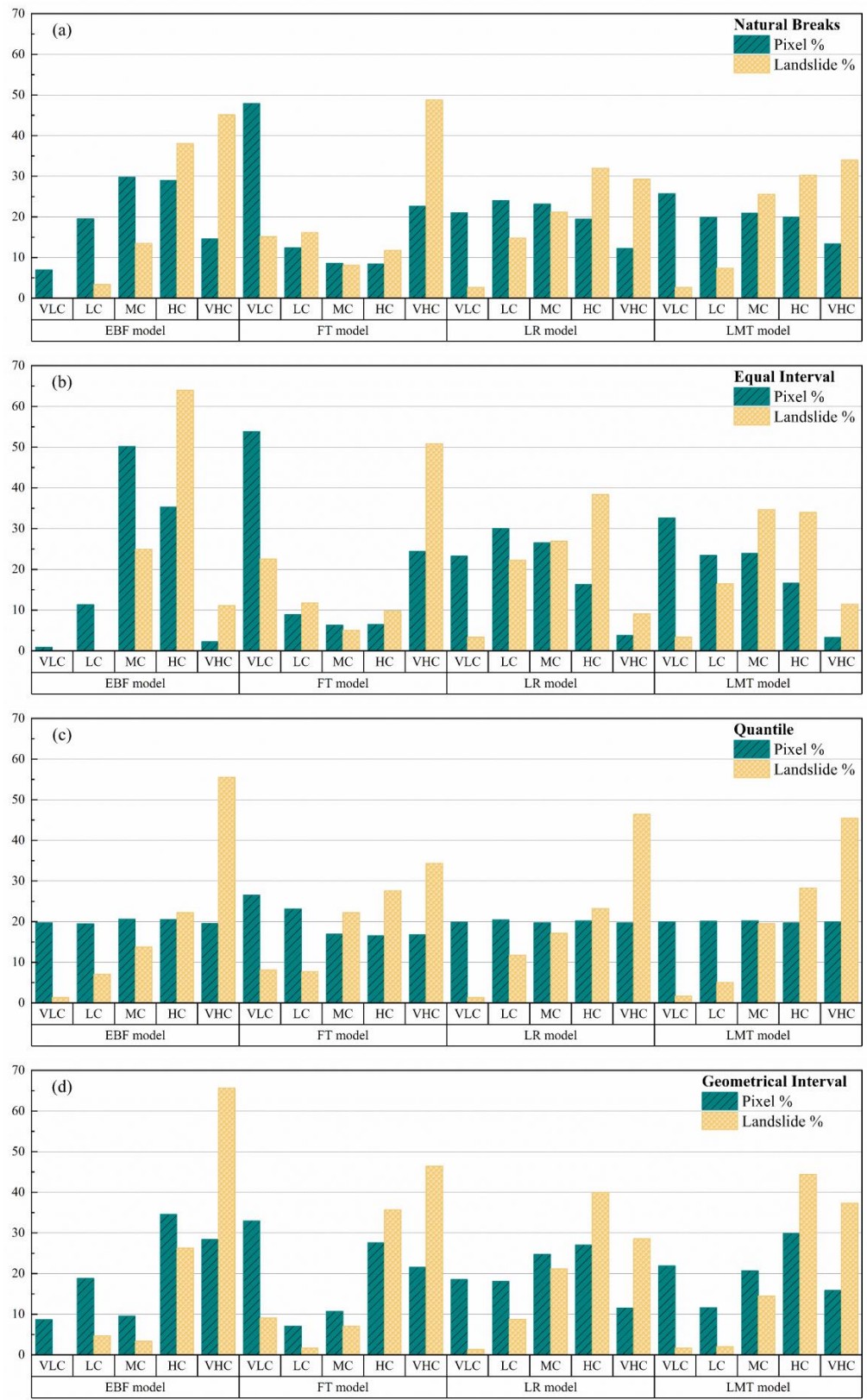

**Figure 13.** Selection of the best classification method for landslide susceptibility maps: (**a**) natural breaks, (**b**) equal interval, (**c**) quantile, and (**d**) geometrical interval.

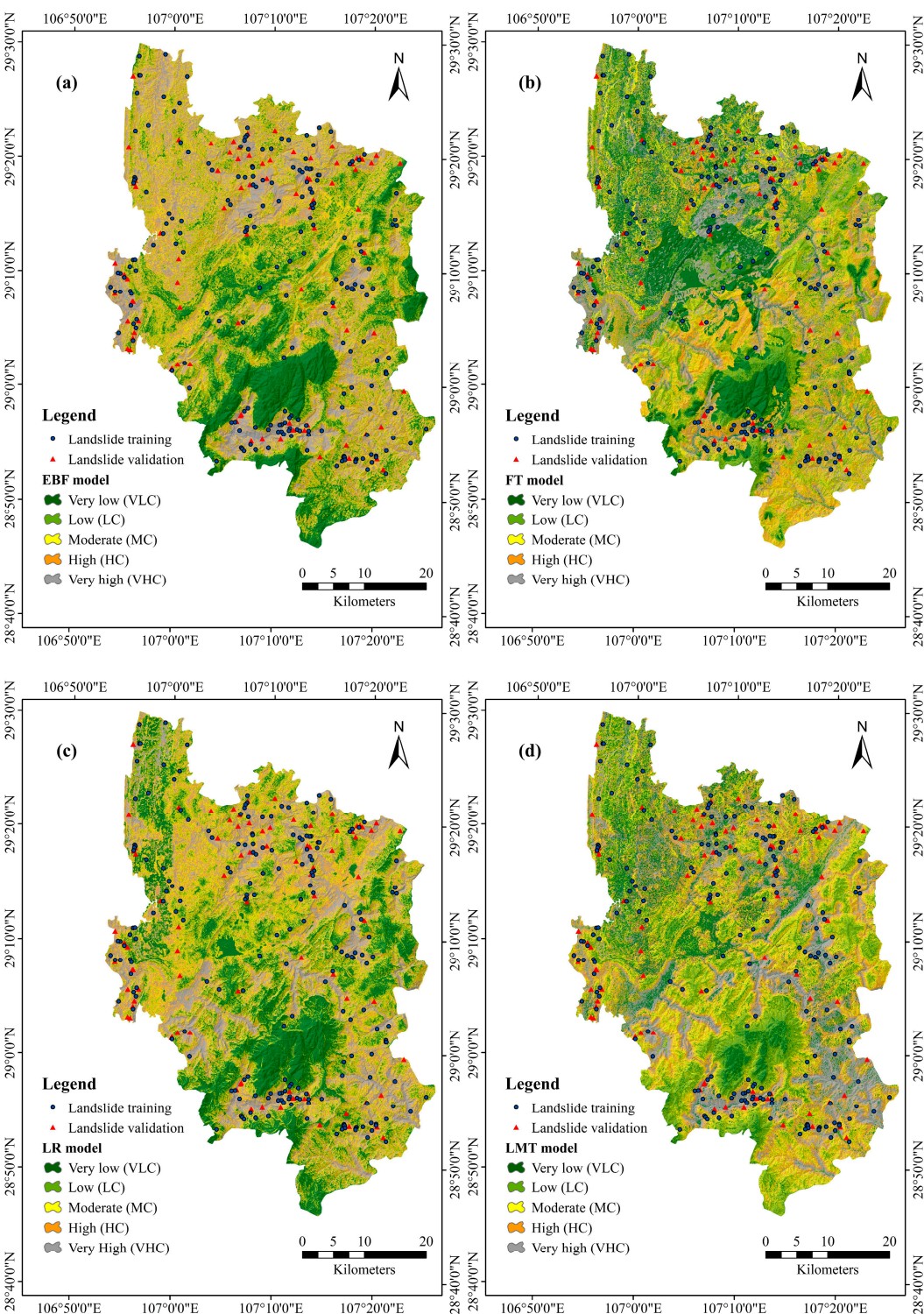

**Figure 14.** Landslide susceptibility maps derived from (**a**) the EBF model, (**b**) the FT model, (**c**) the LR model, and (**d**) the LMT model.

## 6. Discussion

Landslide spatial prediction is an important issue in land use and management [93]. Although different prediction methods exist, these different techniques and methods have the same purpose. In order to obtain accurate and reliable landslide sensitivity prediction results, landslide researchers pay great attention to the establishment of the model. Bivariate algorithms, machine learning algorithms,

and hybrid models are updated on a daily basis. The purpose of this study is to explore the mapping of landslide prone areas in Nanchuan District. In this paper, the evidential belief function (EBF)-based function tree (FT), logistic regression (LR), and logistic model tree (LMT) were applied to Nanchuan District, China. Compared with machine learning models, a bivariate algorithm is not able to achieve satisfactory results in nonlinear modeling [21,96]. Before conducting any research, the two-variable algorithm must define strict assumptions, and the relationship between conditioning factors is largely ignored, that is, the same weight is assumed for different effective factors [21,97]. Furthermore, the internal structure and parameters of the bivariate algorithm are unknown. As a traditional bivariate model, the EBF model is easy to understand and operate, and does not require a complex training process or the adjustment of various parameters. However, EBF models can result in surprises, and the accuracy provided by the EBF model still cannot meet requirements. Machine learning algorithms can improve the prediction ability of the model in the face of complex nonlinear problems. However, the dependence on modeling parameters is high, and the performance of machine learning methods is generally affected by the quality and quantity of training data [45,98]. Training data may also be affected by data distribution, data size, and data resolution [60,99]. Therefore, machine learning algorithms are constantly updated. This paper finds that the FT model is improved by the decision tree model and is more sensitive to the classification of classes. The LR model relies on coefficients to predict binary classification. The LMT model is a classification model composed of the decision tree model and the LR model. It is used as an integrated model to predict and evaluate the susceptibility of landslides in Nanchuan District with the first three models. The ensemble algorithm can be used as a more stable algorithm with higher precision, to produce satisfactory results and improve the prediction ability of the model. By reflecting the global impact and the specific local impact in partitioned data space, the LMT model is ultimately more accurate [100].

Analyzing the performance of the model and optimizing the preform model can ensure the quality of the modeling, establish a robust landslide inventory map, and select 16 landslide conditioning factors from it. The EBF algorithm is used to analyze the correlation between landslide occurrence and the landslide conditioning factors. Rainfall, which has a positive correlation with susceptibility, is the most important factor in this paper, and should be paid attention to. Meanwhile, the influence of other conditioning factors on landslide cannot be ignored. In this research, in order to ensure the quality of conditioning factors, the necessary multicollinearity diagnosis and prediction ability analysis (CAE method) of landslide susceptibility factors are carried out. The results show that there is no interdependency among the 16 selected conditioning factors, and each factor has a different prediction ability and contribution. Therefore, the priority of each factor is determined and applied to the final modeling process.

In this study, the selected models are optimized, the optimal parameters are determined, and excellent parameter configuration is used to ensure the optimization of the model, so as to obtain better model performance. Then, these parameters were chosen to acquire the optimal solution for the landslide susceptibility prediction to provide the best model results. Moreover, the LSI value of each model was calculated and used in the establishment of the landslide susceptibility map. Meanwhile, statistical analysis of the landslide dataset was conducted, and the data distribution, mean standard errors, standard deviation, and variances of each model were obtained. It can be seen from these values that the LMT model was the most accurate for landslide susceptibility evaluation in Nanchuan District, compared to the EBF, FT, and LR models. To find more optimal solutions, it is necessary to apply these models to the study area. According to the ROC curve and AUC value, it is clear that the curve of the LMT model is closest to the upper left corner of the coordinate system, and the AUC values of the training dataset (0.847) and verification dataset (0.765) are also the largest. The FT model has the lowest AUC value among the four models for the training dataset (0.780) and validation dataset (0.676).

In order to undertake a comparative analysis of landslide susceptibility maps among different models, it is necessary to use a variety of classification methods to fully describe the output of these

models. Therefore, in this paper, four classification methods, of Jenks natural breaks, equal interval, quantile, and geometrical interval, were used to divide LSI into five categories: VLC, LC, MC, HC, and VHC. The relative distribution in each category area and the relative proportion of landslides in each category were analyzed, and the quantile classification method was selected to output the classification scheme of landslide susceptibility map output. In this process, not only the rationality of the landslide distribution, but also the rationality between the landslide distribution and the relative distribution in the category area were required. In the landslide susceptibility classification, it can be clearly seen that the susceptibility distribution of the LMT model is superior to the EBF, FT, and LR models (Figure 13c). Therefore, the performance of the LMT model is better than the EBF, LR, and FT models. In short, the integrated algorithm is superior to the single algorithm, and the performance of these models is good. The approach has the correct guiding significance for preventing and controlling future landslides.

## 7. Conclusions

Generally, landslide susceptibility zoning is a useful tool for landslide disaster management and planning. This research was based on four different algorithms (EBF, FT, LR, and LMT) for landslide susceptibility spatial prediction in Nanchuan District. The conditioning factors, landslide datasets, and models were analyzed and improved to achieve the most suitable algorithm. The main results are summarized as follows:

(1) The maps showed that the four landslide susceptibility models were adequate for landslide susceptibility zoning. Compared with the EBF, LR, and FT models, the LMT model showed the best performance.

(2) According to the results of the EBF model, most landslides occur at altitudes of 900–1100 m in the southwest, with a slope angle of 10–20°, plan curvature of −0.05 to 0.05, profile curvature of −27.51 to −0.05, STI > 20, SPI > 20, TWI of 0.24–1, NDVI of 0.20–0.26, farmland category in land use, the fifth group (Ordovician: greyish-black charcoal shale, siliceous base) in lithology, the Dystric Cambisol category in soil, 0–200 m distance to roads, 200–400 m distance to rivers, 1000–2000 m distance to faults, 333.62–1221.86 category in rainfall.

(3) According to the results of the attribute evaluation method, the most factors influencing the occurrence of landslide were the altitude, slope angle, slope aspect, plan curvature, profile curvature, STI, SPI, TWI, NDVI, land use, geological age groups, soil, distance to roads, distance to rivers, distance to faults, and rainfall.

(4) The landslide susceptibility mapping by quantile classification scheme can be a promising tool for government decision makers and engineering technicians.

This method successfully compared four different models and explored the landslide sensitivity in Nanchuan District. The developed method can be used in landslide management and land planning. However, in the future, the ensemble of machine learning techniques for landslide susceptibility modeling still needs to be tested in different cases.

**Author Contributions:** X.Z. and W.C. contributed conceptualization, data selection and preparation, software, method implementation and testing, formal analysis, interpretation of results, preparation of the final manuscript, final approval. All authors have read and agreed to the published version of the manuscript.

**Funding:** This study is supported by the Innovation Capability Support Program of Shaanxi (Program No. 2020KJXX-005).

**Acknowledgments:** Great thanks are given to Xingguang Chen and Zhengqian Wu for their kind help.

**Conflicts of Interest:** The authors declare no conflict of interest.

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
