# Peer review of "Optimization of Computational Intelligence Models for Landslide Susceptibility Evaluation"

_remotesensing, doi:10.3390/rs12142180_

Round 1
Reviewer 1 Report
General comments:
Dear Authors,
the sections “Methods”, “Data preparation” and “Results” were improved.
Nevertheless, more detailed information should be given for the description of the landslide inventory, which plays a fundamental role within this work: which is/are the type/types of landslides included in the inventory? Are Authors dealing with an either homogeneous or heterogeneous inventory (classification, size, state of activity, etc.)? This is very important to highlight, because different types of landslide may be related to different causal factors or different combinations of the same factors.
The “Study area” and “Discussions” sections are not developed and detailed enough, still requiring major improvement.
More specific comments are given in the following.
Best regards.
Specific comments:
Pag. 5 “Study Area”
A description of the geological setting of the area is still completely missing. It would be useful to use a map, instead of describing the drainage network. Moreover, the terms used to describe the drainage network appear to be inappropriate and more strict reference to the relevant literature should be done for specific terms.
Pag. 6 “In this study, 298 landslides were identified by using a literature search. By referring to the historic reports, Google Earth satellite image interpretations, and through field investigation, landslide areas were converted into point data (landslide centroids).”
Again, the general picture is rather unclear for the reader. Which are the reports sources and related literature? What about their accuracy? What about the accuracy of results of remote imagery interpretation? What about the classification (ex. Varnes, Hungr, Hutchinson), size, state of activity? Does the paper deals with either shallow landslides mainly involving unconsolidated slope deposits and weak rocks, or deep landslides mainly involving bedrock units? Authors should carefully highlight and describe these aspects because the causal factors affect these two groups of landslides following different rules. Hence, the susceptibility analysis should be performed independently for the different datasets of landslides. Anyway, no information is given about the above issues in the manuscript.
I underline that the significance of this paper strongly depends on the significance of the landside inventory!
Pag. 6 “Digital elevation model (DEM) was extracted from Aster GDEM data (http://www.gscloud.cn). DEM was utilized to extract thematic data layers such as elevation, slope angle, aspect, plan curvature, profile curvature, STI, SPI and TWI. Landsat 8 OLI images, traffic map, and 1:200,000 geological maps were used to extract the NDVI, distance to roads, distance to rivers, and distance to faults. Using the local rainfall data, the rainfall map was drawn based on the inverse distance weighted method [62,63]. The landuse thematic data were drawn from the 1:100,000 landuse map. Meanwhile, the soil thematic data were extracted by using the 1:1,000,000 scale soil map.”
The geological map should be represented in the manuscript. Only lithology is represented in Fig. 3, hence no information is given to the reader about faults distribution, which is a set of information used to analyse landslide susceptibility. Does a more detailed scale than 1:200,000 exist for geology? Please, also: a) the code/software used to obtain the raster outputs from the DEM should be reported; b) the map of rainfall gauges (chapter study area) should be added.
Pag. 7 “Geological factors” Lithology
Looking at the table it seems that lithology classes were obtained by grouping different lithological units on the base of the age, not really the lithology!!!!! Therefore, class 1 Jurassic includes mudstone and limestone together. But these two types have a different behaviour in respect to their tendency to develop landslides. The Authors should add more geological information and a geological map in order to support and justify the reasons of the way of lithological grouping. If deciding to maintain the same grouping rules they shouldn’t name these groups lithology groups, but eventually “geological age groups”…
Figure 3.
I'm sorry, but both the curvature maps continue to appear very strange to me. No chromatic variability may be recognized within these maps.
Figure 5.
Rows of factors are not aligned to light blue bars. Why? Is it a mistake? Please check!
Pag. 9-10 “Regarding the slope angle, the Bel value of the southwest slope (0.148) is the largest, and that of the flat is zero”
Unclear to the reader. Do you mean slope aspect? The authors are referring to “southwest”… Are you referring to slope aspect?
Pag. 17 “In this research……., ……, which were transformed into ArcGIS to generate the ultimate map.”
Why are Authors talking about scholars? What is the meaning they are giving to this term? Natural breaks approach is an intuitive and fast method that classifies based on the values distribution. However, by using this approach each susceptibility model will have its own values distribution depending on the parameters used to make calculations. Since the goal of the work is to perform comparison among different methods, it is necessary to use one or more classification methods that may be adequate to describe all the model’s output (as an example: fixed quantiles).
Another way to give more reliable susceptibility classes could be to analyse the slope of the ROC curve, as already known in the literature.
Pag. 19-20 “Discussion”
“In the slope angle, although the Bel value was the largest (0.330) in the range of 60–70° with a regional proportion of 0.330, the number of landslides is the largest in the range of 10–20° with a larger area (0.164).”
Not clear to the reader. Do Authors intend to observe that landside frequency is higher in the area in the range 10°-20° and this is the widest area? If this is the case, isn’t it “as expected”? Why not? Didn’t they perform any area-based normalization of the data? Please explain.
“Regarding geology, such areas in the range of 10–20° are most prone to landslides due to its instability after heavy rain [81].”
Not clear to the reader. Which are the relationships between geology and slope steepness? Where did Authors discuss this?
“Many experts and scholars have identified that shale transports permeated water to the fracture surface [88].”
Not clear to the reader. What is the meaning of this sentence? Please explain and improve the text.
“In addition, the linear features of distance to rivers, distance to faults, and distance to roads are not clear in this study. But the occurrence of landslides and these factors are inseparable..”
Not clear to the reader also! What is the meaning of “inseparable”? Why are the effects of these parameters not clear? Please explain and improve the text.
“As is known, the two variable algorithms cannot obtain satisfactory results when dealing with very complex nonlinear problems.”
Sentences like this shouldn’t be used. Authors have to discuss and demonstrate/support here, based on the relevant literature, why they affirm this concept. Please, improve the text.
In a general perspective, this chapter is generally poor, and some sentences should be moved to the Results sections (ex.: the description of the values indicating correlation with landslides occurrence).
The topics of the paper are not adequately explored. As an example, it would be interesting for the readers a critical discussion about the distribution of landslides within the classes obtained by the different approaches.
The Authors should also explain how to interpret the results shown within "Figure 12". Even if the LMT model have the better AUROC value, almost 50% of the landslides fall in the “very low” and low” susceptibility classes. Could the Authors explain: a) why is this happening? b) why the results of three methods over four provide results where frequency of landslides is higher in the low susceptibility classes instead of the opposite? c) having chosen natural breaks approach to obtain classes, do the Authors think that classes by the different methods are each other comparable? Is opinion of the reader that they are not! d) are the results acceptable and selective? e) how the quality of the landslide inventory could influence the quality of the susceptibility assessment results?
Author Response
Dear Editor-in-Chief Many thanks for useful comments and suggestions on our manuscript entitled "Optimization of computational intelligence models for landslide susceptibility evaluation" (remotesensing-824363). The suggestions are quite helpful for improving the MS and we incorporate them into the revised manuscript. We have addressed all concerns as outlined below. We have made use of the language editing service provided by MDPI (with a certificate). Finally, we hope that the revised manuscript can be acceptable in its present form. Thanking you in advance. Yours Sincerely, List of changes in the revised paper: This document explains the changes made in the revised manuscript while dealing with the comments raised by the reviewers. Reviewer’s comments are marked in black; author’s response is shown in blue, while the changes in the manuscript are marked in red. COMMENTS FROM EDITORS AND REVIEWERS Peer Reviewer #1: Dear Authors, the sections “Methods”, “Data preparation” and “Results” were improved. Nevertheless, more detailed information should be given for the description of the landslide inventory, which plays a fundamental role within this work: which is/are the type/types of landslides included in the inventory? Are Authors dealing with an either homogeneous or heterogeneous inventory (classification, size, state of activity, etc.)? This is very important to highlight, because different types of landslide may be related to different causal factors or different combinations of the same factors. The “Study area” and “Discussions” sections are not developed and detailed enough, still requiring major improvement. Response: Dear Reviewer #1, thank you very much for your valuable comments on our manuscript. We have responded to your comments point to point, mainly to add relevant content from the study area, landslide inventory and discussion. The whole manuscript has been proofread by MDPI (https://www.mdpi.com/authors/english ), you can see its certificate. Thank you for giving me this opportunity to introduce our views on this article. More specific comments are given in the following. Specific comments: Comment 1: Pag. 5 “Study Area” A description of the geological setting of the area is still completely missing. It would be useful to use a map, instead of describing the drainage network. Moreover, the terms used to describe the drainage network appear to be inappropriate and more strict reference to the relevant literature should be done for specific terms. Response: Agree and change made. Thank you for your good suggestion. The whole manuscript has been proofread by MDPI (https://www.mdpi.com/authors/english ), you can see its certificate. Nanchuan District is part of Chongqing, China, and is located at 106° 54′–107° 27′ E and 28° 46′–29° 30′ N. Nanchuan District is adjacent to Wulong in the northeast, Guizhou in the southeast, Fuling in the north, and Banan, Qijiang and Wansheng in the west. It is the necessary transportation gateway of Southern Chongqing and Northern Guizhou. The district is 80.25 km in length from south to north, 52.5 km wide from east to west, and has a total area of 2601.92 km2 (Figure 1). Furthermore, Nanchuan District is located in a subtropical warm monsoon area with abundant rainfall and a mild climate. The annual average temperature is 20 °C. The maximum annual rainfall is 1534.80 mm, and the annual average rainfall is 1170.20 mm. The seasonal distribution of atmospheric precipitation is uneven, the driest month is January, and the rainy season is mainly from May to September (http://www.weather.com.cn/). Figure 1. The study area. Nanchuan District is located on the southeastern margin of the Sichuan Basin and to the northwest of the Dalou Mountains. The area has the characteristics of strong settlement in the northwest (within the basin) and uplift in the southeast. The lithology in the area is mainly mudstone, sandstone, and limestone, and Quaternary deposits is widely distributed in depressions, river valleys, and slopes. Paleozoic and Mesozoic deposits are widely distributed in the area. The lithology is mainly carbonate rocks and clastic rocks, and there are a few Quaternary loose accumulation layers, which laid the foundation for the formation of groundwater and constituted carbon. The three basic types of groundwater are karst water in salt rock, fissure water in bedrock, and pore water in loose rock. The surface water system in the area mainly comprises the Yangtze River system, which is mostly branched, followed by feathers, with a large slope drop and rapid water flow. Comment 2: Pag. 6 “In this study, 298 landslides were identified by using a literature search. By referring to the historic reports, Google Earth satellite image interpretations, and through field investigation, landslide areas were converted into point data (landslide centroids).” Again, the general picture is rather unclear for the reader. Which are the reports sources and related literature? What about their accuracy? What about the accuracy of results of remote imagery interpretation? What about the classification (ex. Varnes, Hungr, Hutchinson), size, state of activity? Does the paper deals with either shallow landslides mainly involving unconsolidated slope deposits and weak rocks, or deep landslides mainly involving bedrock units? Authors should carefully highlight and describe these aspects because the causal factors affect these two groups of landslides following different rules. Hence, the susceptibility analysis should be performed independently for the different datasets of landslides. Anyway, no information is given about the above issues in the manuscript. I underline that the significance of this paper strongly depends on the significance of the landside inventory! Response: Agree and change made. Thank you for your good suggestion. Obtaining and constructing datasets of the landslide conditioning factors and the landslide inventory is necessary for landslide susceptibility zoning and analysis [58]. The quality of the landslide inventory determines the results of landslide susceptibility prediction and evaluation. However, there is currently no detailed standard for the accuracy of landslide inventory [59]. In the overall process of landslide susceptibility prediction, based on statistics, it is assumed that the conditions of future landslides are the same as those of the past [16,60]. In this study, 298 landslides were identified based on field investigations, historical reports, and Google Earth satellite image interpretations. A complete landslide inventory map of Nanchuan District was used to identify and record the location (centroid) of these previous landslides, which consists of 296 slides and 3 rockfalls (Figure 1). This study uses landslide events caused mainly by multiple rainfalls during the time span of 1979 to 2018. According to the analysis of landslides using GIS tools, the smallest landslide area was close to 70 m2, the largest was more than 8.4×105 m2, and the average area was about 3.07×104 m2. In terms of volume, the minimum volume was only 140 m3, and more than 95% of the landslides were less than 100,000 m3. The occurrence of landslides is closely related to the exposed strata, and its lithological conditions are the decisive factors that determine landslides’ occurrence. Landslides are prone to occur in strata distribution areas such as clay, mudstone, shale, and marl. Based on the above, it can be shown that the established landslide inventory map is sufficiently robust and can be used for the landslide susceptibility analysis in this study. The split of the dataset is of significance to verify the performance of the model [61]. Furthermore, through the analysis and comparison of landslide data, it was found that 70%:30% could be used as the classification standard of landslide data in this paper [62-64]. In addition, an equal amount (298) of non-landslide points were randomly selected in areas without landslides, and then allocated 70%/30% to the training and verification data sets, respectively. Then, data were assigned values of 1 (with landslide) and 0 (without landslide). Comment 3: Pag. 6 “Digital elevation model (DEM) was extracted from Aster GDEM data (http://www.gscloud.cn). DEM was utilized to extract thematic data layers such as elevation, slope angle, aspect, plan curvature, profile curvature, STI, SPI and TWI. Landsat 8 OLI images, traffic map, and 1:200,000 geological maps were used to extract the NDVI, distance to roads, distance to rivers, and distance to faults. Using the local rainfall data, the rainfall map was drawn based on the inverse distance weighted method [62,63]. The landuse thematic data were drawn from the 1:100,000 landuse map. Meanwhile, the soil thematic data were extracted by using the 1:1,000,000 scale soil map.” The geological map should be represented in the manuscript. Only lithology is represented in Fig. 3, hence no information is given to the reader about faults distribution, which is a set of information used to analyse landslide susceptibility. Does a more detailed scale than 1:200,000 exist for geology? Please, also: a) the code/software used to obtain the raster outputs from the DEM should be reported; b) the map of rainfall gauges (chapter study area) should be added. Response: Agree and change made. Thank you for your good suggestion. We have added fault information to Figure 4(k) in the study area. This figure can also be used as a geological map in this article. Based on our current circumstance, we can only obtain 1:200,000 geological map in the study area, and a more detailed scale is in the direction of our future efforts. This paper uses ArcGIS 10.5 software to obtain raster data outputs from the DEM. For the rainfall map, we used 112 rain gauges to map the rainfall map, however, the locations of these rainfall gauges are not allowed to be shown in the published paper. I hope we can get your understanding. Thank you for your advice and give me the chance to express my opinion. Figure 3-1. geological map in the study area Figure 3-2. Rainfall station data in the study area Figure 3-3. Rainfall gauges map in the study area Comment 4: Pag. 7 “Geological factors” Lithology Looking at the table it seems that lithology classes were obtained by grouping different lithological units on the base of the age, not really the lithology!!!!! Therefore, class 1 Jurassic includes mudstone and limestone together. But these two types have a different behaviour in respect to their tendency to develop landslides. The Authors should add more geological information and a geological map in order to support and justify the reasons of the way of lithological grouping. If deciding to maintain the same grouping rules they shouldn’t name these groups lithology groups, but eventually “geological age groups”… Response: Agree and change made. Thank you for your good suggestion. We have replaced “lithology” with “geological age groups” and added some geological data information in the section of description of the study area. Hope to get your approval. Comment 5: Figure 3. I'm sorry, but both the curvature maps continue to appear very strange to me. No chromatic variability may be recognized within these maps. Response: I agree with you that the color difference between the two maps is not obvious. However, due to the large gap between the curvature values of adjacent locations and the wide distribution of similar curvature values, it is difficult for the map to morphologically obtain a map with distinctive color differences. We tried different classification methods, but still got similar maps, so in order to maintain the consistency of this set of map formats, we can only choose this set of maps to continue to use. Thank you very much for your valuable comments and give us the opportunity to express our views. Comment 6: Figure 5. Rows of factors are not aligned to light blue bars. Why? Is it a mistake? Please check! Response: Agree and change made. Thank you for your good suggestion. We have not listed the specific AM and its standard deviation here, so it is difficult to understand the existence. The standard deviation of slope angle and altitude is 0, so it is not shown in the figure. Now, on the premise that the data is accurate and reliable, and the error does not affect the prediction and comparison of various factors, the upper and lower limits of the standard deviation are removed. Figure 5. The prediction ability of the landslide conditioning factors. Comment 7: Pag. 9-10 “Regarding the slope angle, the Bel value of the southwest slope (0.148) is the largest, and that of the flat is zero” Unclear to the reader. Do you mean slope aspect? The authors are referring to “southwest”… Are you referring to slope aspect? Response: Agree and change made. Thank you for your good suggestion. Due to my carelessness, this mistake has been corrected, so it should be slope aspect. Regarding the slope aspect (Figure 4(c)), the Bel value of the southwest (0.148) is the largest, and that of the flat is zero. Comment 8: Pag. 17 “In this research……., ……, which were transformed into ArcGIS to generate the ultimate map.” Why are Authors talking about scholars? What is the meaning they are giving to this term? Natural breaks approach is an intuitive and fast method that classifies based on the values distribution. However, by using this approach each susceptibility model will have its own values distribution depending on the parameters used to make calculations. Since the goal of the work is to perform comparison among different methods, it is necessary to use one or more classification methods that may be adequate to describe all the model’s output (as an example: fixed quantiles). Another way to give more reliable susceptibility classes could be to analyse the slope of the ROC curve, as already known in the literature. Response: Agree and change made. Thank you for your good suggestion. In order to illustrate the classification of LSI, we use the classification methods proposed by some scholars, because the writer expresses their apologies improperly and correct them. According to your suggestion and the optimization of the classification method, it is finally determined that the quantile you said is the most suitable LSM classification method for this study, and the specific content has been added to the paper. In this study, a unique LSI was applied to all pixels in this research area to establish a landslide susceptibility model. The classification methods of LSI mainly include Jenks natural breaks, quantile, geometrical interval, equal interval, and standard deviation [92,93]. In this study, the four classification methods of Jenks natural breaks, equal interval, quantile, and geometrical interval were used to divide LSI into five categories: very low (VLC), low (LC), moderate [54], high (HC) and very high (VHC). The relative distribution of susceptibility category areas and the relative proportion of landslides in each category in the study are shown in Figure 11. Generally, the histograms of different models under different classification methods show regularity: the higher the susceptibility category, the greater the landslide distribution. Most landslides were recorded in the very high (VHC) category. It can be clearly seen from Figure 11 that this rule exists, and the quantile is superior to the other three classification methods, with outstanding performance. In the FT model, the quantile classification method does not perform as well as the other three models, but for the FT model, the quantile classification method is still its best choice. Therefore, the quantile is used as a classification scheme for landslide susceptibility maps of the EBF, FT, LR, and LMT models. Figure 11. Selection of the best classification method for landslide susceptibility maps: (a) natural breaks, (b) equal interval, (c) quantile, and (d) geometrical interval. The final four landslide susceptibility maps were drawn according to the selected classification method (Figure 12). In the EBF model (Figure 12(a)), the scales of the VLC (19.75%), LC (19.51%), MC (20.65%), HC (20.51%), and VHC (19.59%) classifications are distributed. The proportions of landslides in the areas of various categories are VLC (1.35%), LC (7.07%), MC (13.80%), HC (22.22%), and VHC (55.56%). In the modeling results of the FT model (Figure 12(b)), the proportions of VLC (26.52%), LC (23.14%), MC (16.94%), HC (16.58%) and VHC (16.82%) categories are distributed. The proportions of landslides in the areas of each category are VLC (8.08%), LC (7.74%), MC (22.22%), HC (27.61%), and VHC (34.34%). According to the landslide susceptibility zoning results of the LR model (Figure 12(c)), the scales of the VLC (19.86%), LC (20.47%), MC (19.77%), HC (20.20%), and VHC (19.70%) classifications are distributed. The proportions of landslides in the areas of each category are VLC (1.35%), LC (11.78%), MC (17.17%), HC (23.23%), and VHC (46.46%). Finally, in the modeling results of the LMT model (Figure 12(d)), the proportions of VLC (20.00%), LC (20.11%), MC (20.24%), HC (19.69%), and VHC (19.96%) categories are distributed. The proportions of landslides in the areas of each category are VLC (1.68%), LC (5.05%), MC (19.53%), HC (28.28%), and VHC (45.45%). Figure 12. Landslide susceptibility maps derived from (a) EBF model, (b) FT model, (c) LR model, and (d) LMT model. Comment 9: Pag. 19-20 “Discussion” “In the slope angle, although the Bel value was the largest (0.330) in the range of 60–70° with a regional proportion of 0.330, the number of landslides is the largest in the range of 10–20° with a larger area (0.164).” Not clear to the reader. Do Authors intend to observe that landside frequency is higher in the area in the range 10°-20° and this is the widest area? If this is the case, isn’t it “as expected”? Why not? Didn’t they perform any area-based normalization of the data? Please explain. Response: Agree and change made. Thank you for your good suggestion. Here is to explain why the Bel value we obtained is in the range of 60-70° (accounting for 0.17% of the total area) greater than the range of 10-20° (accounting for 35.95% of the total area). In terms of slope angle, although the Bel value is the largest in the range of 60-70° (0.330), the landslides in the range account for only 0.47% of the total, while the landslides in the range of 10-20° account for 49.28% of the total, Bel value is 0.164. According to the data, this does not imply that there are special circumstances but is in line with expectations. Thank you very much for your good suggestions and give us the opportunity to elaborate our views. In the eight categories classified according to the slope angle (Figure 4(b)), although the Bel value is the largest in the range of 60–70° (0.330), this range only accounts for 0.17% of the total area, and landslide in this range only account for 0.47% of the total. The Bel value is only 0.164 in the range of 10–20°, accounting for 35.95% of the total area. None landslides are found when the slope angle is >70°. According to the data, this does not imply that there are special circumstances, but is in line with expectations. Regarding geology, regions of the range of 10–20° are most prone to landslides due to their instability after heavy rain [70]. Comment 10: “Regarding geology, such areas in the range of 10–20° are most prone to landslides due to its instability after heavy rain [81].” Not clear to the reader. Which are the relationships between geology and slope steepness? Where did Authors discuss this? Response: What this sentence wants to express is that our results are consistent with the opinions of the cited documents. From the geological point of view, the stability of the slope is changed by the impact of heavy rain, and the area with a slope of 10-20 ° is more prone to landslide disaster. Thank you very much for your good comments and give us the opportunity to present our views. Comment 11: “Many experts and scholars have identified that shale transports permeated water to the fracture surface [88].” Not clear to the reader. What is the meaning of this sentence? Please explain and improve the text. Response: Agree and change made. Thank you for your good suggestion. The whole manuscript has been proofread by MDPI (https://www.mdpi.com/authors/english ), you can see its certificate. Thank you for giving me this opportunity to introduce our views on this article. Comment 12: “In addition, the linear features of distance to rivers, distance to faults, and distance to roads are not clear in this study. But the occurrence of landslides and these factors are inseparable.” Not clear to the reader also! What is the meaning of “inseparable”? Why are the effects of these parameters not clear? Please explain and improve the text. Response: What I want to express here is that the linear characteristics of river distance, fault distance and road distance in this study are not prominent, and their respective regular pattern are not significant, but these landslide conditioning factors have a certain impact on the occurrence of landslides. Thank you very much for your good suggestions and give us the opportunity to elaborate our views. Comment 13: “As is known, the two variable algorithms cannot obtain satisfactory results when dealing with very complex nonlinear problems.” Sentences like this shouldn’t be used. Authors have to discuss and demonstrate/support here, based on the relevant literature, why they affirm this concept. Please, improve the text. Response: Agree and change made. Thank you for your good suggestion. Compared with machine learning models, a bivariate algorithm is not able to achieve satisfactory results in nonlinear modeling [21,95]. Before conducting any research, the two-variable algorithm must define strict assumptions and the relationship between conditioning factors is largely ignored, that is, the same weight is assumed for different effective factors [21,96]. Furthermore, the internal structure and parameters of the bivariate algorithm are unknown. As a traditional bivariate model, the EBF model is easy to understand and operate, and does not require a complex training process or adjustment of various parameters. However, EBF models can result in surprises and the accuracy provided by the EBF model still cannot meet requirements. In a general perspective, this chapter is generally poor, and some sentences should be moved to the Results sections (ex.: the description of the values indicating correlation with landslides occurrence). The topics of the paper are not adequately explored. As an example, it would be interesting for the readers a critical discussion about the distribution of landslides within the classes obtained by the different approaches. The Authors should also explain how to interpret the results shown within "Figure 12". Even if the LMT model have the better AUROC value, almost 50% of the landslides fall in the “very low” and low” susceptibility classes. Could the Authors explain: a) why is this happening? b) why the results of three methods over four provide results where frequency of landslides is higher in the low susceptibility classes instead of the opposite? c) having chosen natural breaks approach to obtain classes, do the Authors think that classes by the different methods are each other comparable? Is opinion of the reader that they are not! d) are the results acceptable and selective? e) how the quality of the landslide inventory could influence the quality of the susceptibility assessment results? Response: Agree and change made. Thank you for your good suggestion. We have modified the discussion substantially and moved the numerical description related to landslide occurrence to the results section. We add the different classification methods mentioned by you to analyze the distribution of landslides within the classes obtained by the different approaches and hope to get your approval. Through discussion, we find that the quantile, as the most suitable classification scheme for this paper, completely shows our classification of landslide sensitivity, and the law between classes is very reasonable. In our opinion the manuscript deserves an opportunity to be evaluated for the following reasons: 1. As we all know, landslide is a very complex natural phenomenon, which has caused serious casualties and property losses all over the world. The occurrence of these extreme events needs to be accurately evaluated in order to understand their spatial correlation with landslides. An effective method is to map the prone area of landslide. In recent years, various machine learning techniques have been applied to landslide susceptibility mapping. However, we cannot determine which model is the most common. In addition, even if the prediction accuracy is slightly improved, the landslide prone zone may also be controlled. Therefore, more case studies are needed to draw reasonable conclusions. For this matter, we can see that many researchers have published papers using various data and models. At the same time, we are also working on this. 2. This method combines EBF-based FT, LR and LMT. Modeling these different processes is a great challenge and is considered the right solution to overcome the problems that arise when each method is implemented separately. In particular, these major new procedures are introduced in the current study, which is different from other studies. Firstly, the spatial correlation between the condition factor and the landslide is analyzed by using the two variable EBF model. Secondly, the optimization of FT, LR and LMT models based on EBF is implemented, and a reasonable classification method is selected to construct landslide susceptibility map. 3. The proposed approach is an innovative method that may also help other scientists to develop landslide susceptibility maps in other areas but also as an approach that could be used in geo-environmental problems besides natural hazard assessments. 4. We checked the manuscript several times to eliminate typographical and typographical errors, including forms and numbers. And MDPI proofread the whole manuscript (https://www.mdpi.com/authors/english ). Once again, I would like to thank you for giving me this opportunity to present our views. Hope to get your approval. Waiting for your positive response.

Reviewer 2 Report
This paper compares three methods of landslide susceptibility mapping. It is essential to prepare and compare landslide susceptibility maps of landslide-prone areas using various methods. In this regard, this research has scientific values. The data used in this research are based on remote sensing; therefore, it fits the scope of this journal. But this paper has several issues that needed to be revised. Details are given in the attached text.

Author Response
Dear Editor-in-Chief
Many thanks for useful comments and suggestions on our manuscript entitled "Optimization of computational intelligence models for landslide susceptibility evaluation" (remotesensing-824363). The suggestions are quite helpful for improving the MS and we incorporate them into the revised manuscript. We have addressed all concerns as outlined below. We have made use of the language editing service provided by MDPI (with a certificate). Finally, we hope that the revised manuscript can be acceptable in its present form.
Thanking you in advance.
Yours Sincerely,
List of changes in the revised paper:
This document explains the changes made in the revised manuscript while dealing with the comments raised by the reviewers. Reviewer’s comments are marked in black; author’s response is shown in blue, while the changes in the manuscript are marked in red.
COMMENTS FROM EDITORS AND REVIEWERS
Peer Reviewer #2:
This paper compares three methods of landslide susceptibility mapping. It is essential to prepare and compare landslide susceptibility maps of landslide-prone areas using various methods. In this regard, this research has scientific values. The data used in this research are based on remote sensing; therefore, it fits the scope of this journal. But this paper has several issues that needed to be revised. Details are given in the attached text.
Response: Dear Reviewer #2, thank you very much for your valuable comments on our manuscript. At the same time, thank you for modifying the content and format of the manuscript. Based on your revised version, we responded to your comments one by one. And because of the language problem, the whole manuscript has been proofread by MDPI (https://www.mdpi.com/authors/english ), you can see its certificate. Thank you for giving me this opportunity to present our views on this paper.
- Please mention the name of the study area.
Response: Agree and change made. Thank you for your good suggestion.
This paper focuses on landslide susceptibility prediction in Nanchuan, a high-risk landslide disaster area.
- Mention the type of accuracy like success and prediction rate.
Response: Agree and change made. Thank you for your good suggestion.
In particular, the LMT model had the best performance (0.847 and 0.765, obtained from the training and validation dataset, respectively).
- Is it a good key word?
Response: Agree and after much deliberation, we decided to delete the bad keyword. Thank you for your good advice.
Keywords: Evidential Belief Function; Function Tree; Logistic Regression; Logistic Model Tree; Nanchuan District.
- Language is very hard to understand.
Response: Agree and change made. Thank you for your good suggestion. The whole manuscript has been proofread by MDPI (https://www.mdpi.com/authors/english ), you can see its certificate.
Landslides occur mainly due to the landform, geology, hydrology, soil, meteorology, human activities, and land-use patterns of the region under different geospatial and geographic conditions [1,2]. Landslides often represent a danger to human beings and cause property damage due to natural conditions and human engineering activities [3].
- It is kind of repetition. You have mentioned that landslide cause death of life in previous sentence. We can anticipate that these landslides in China has caused some sort of damage.
Response: Agree and change made. Thank you for your good suggestion. The whole manuscript has been proofread by MDPI (https://www.mdpi.com/authors/english ), you can see its certificate.
In recent years, many areas of China have suffered from landslides, posing a clear threat to the safety of communication equipment, transmission lines, and transportation networks [4].
- What does it mean? --“The selection of appropriate analysis, … ”
Response: This refers to the landslide susceptibility prediction, we need to choose the appropriate method to analyze, when the analysis method is selected, it will greatly reduce our analysis workload, improve prediction accuracy and reliability, and improve the prediction process. Here, we summarize the necessary theoretical basis for landslide susceptibility modeling based on the cited literature. Thank you for giving me the opportunity to present my views.
- Or quantitative?? --“Generally, a qualitative method uses a landslide inventory to ... ”
Response: The proposed qualitative method is compared with the subsequently proposed quantitative method from a functional perspective. The subjectivity of the qualitative method based on expert knowledge is proposed, which proves the practicability of the quantitative method selected in this paper. Thank you for giving me the opportunity to present my views.
- Please cite some articles here.
Response: Agree and change made. Thank you for your good suggestion.
Because the qualitative method based on expert knowledge may be too subjective, the quantitative approach is increasingly being applied to landslide susceptibility modeling [13,15,16].
- Is it a good word? --“…, statistically-based methods, … ”
- Response: Agree and change made. Thank you for your good suggestion.
It’s replaced with “statistical-based methods”. Besides, the whole manuscript has been proofread by MDPI (https://www.mdpi.com/authors/english ), you can see its certificate.
- What about logistic regression? --“…a specific hypothesis and over require parameters.”
Response: The role of logistic regression in the text is to compare and analyze with the other three models as a representative of this type of statistical analysis. In this paper, we also analyze the parameters of logistic regression and list the main expressions (Eq. 12). Thank you for giving me the opportunity to present my views.
- What is it? --“… produce more reliable results through data statistics.”
Response: What I want to express here is the inevitable connection between data statistics and machine learning methods. As we all know, machine learning models are outstanding in mathematics (Zhou et al., 2018). Among data-driven models, machine learning models are considered more effective because of their mathematical expertise (Goetz et al., 2015; Pham et al., 2016). The requirements for data quality are getting higher and higher, and a good set of data is more conducive to producing good results on the regional landslide susceptibility model. In the article, only the advantages of this point are explained. Thank you for giving me the opportunity to present my views.
Zhou, C.; Yin, K.; Cao, Y.; Ahmed, B.; Li, Y.; Catani, F.; Pourghasemi, H.R. Landslide susceptibility modeling applying machine learning methods: A case study from longju in the three gorges reservoir area, china. Comput. Geosci. 2018, 112, 23-37.
Goetz, J.N.; Brenning, A.; Petschko, H.; Leopold, P. Evaluating machine learning and statistical prediction techniques for landslide susceptibility modeling. Comput. Geosci. 2015, 81, 1-11.
Pham, B.T, Bui, D.T, Dholakia, M.B., Prakash, I., Pham, H.V, Mehmood, K, Le, H.Q. A novel ensemble classifier of rotation forest and Naive Bayer for landslide susceptibility assessment at the Luc Yen district, Yen Bai Province (Viet Nam) using GIS. Geomatics, Nat. Hazards Risk, 2016, 1-23.
- So, machine learning method is more useful. For example: authors should provide why it is useful. More application does not make it more useful.
Response: Agree and change made. Thank you for your good suggestion. The reason why the machine learning method is more useful is because this is highly recognized by experts and scholars, and some articles are cited in the manuscript to illustrate this point. Of course, this is only general, not comprehensive. Not every model has such advantages, and the accuracy that each model can obtain for the same data is also different. Pourghasemi and Rahmati conducted a comparative study on the ten most advanced machine learning models selected. Thank you for giving me the opportunity to present my views.
Pourghasemi, H.R.; Rahmati, O. Prediction of the landslide susceptibility: Which algorithm, which precision? Catena 2017, 162, 177-192.
- Use easy words
Response: Agree and change made. Thank you for your good suggestion. The whole manuscript has been proofread by MDPI (https://www.mdpi.com/authors/english ), you can see its certificate.
It can be clearly seen from the literature review that additional comparative studies of distinct machine learning models are required for better problem analysis, evaluating the effectiveness of models, and to help improve landslide susceptibility prediction [19,34,44].
- Change this sentence. Try to make it easily understandable.
Response: Agree and change made. Thank you for your good suggestion. The whole manuscript has been proofread by MDPI (https://www.mdpi.com/authors/english ), you can see its certificate.
This paper is mainly divided into five parts, as shown in Figure 2.
- Please provide more explanations.
Response: Agree and change made. Thank you for your good suggestion.
The EBF is primarily based on the Dempster–Shafer evidence algorithm theory [45,46]. Dempster–Shafer theory, as an extension of Bayesian subjective probability theory, mainly deals with the influence of the confidence degree of the problem and the probability of the problem.
The main advantage of applying this method to a landslide susceptibility study is its adaptability. This is mainly due to the integration of beliefs from multiple sources and the acceptability of uncertainty. The other advantage of the EBF model is that it is a method of uncertainty prediction in the landslide mapping area [25]. These advantages lead to the EBF model having good prediction results as a two-variable model. Belief (Bel) multi-tier integration is represented by the following formula [47]:
where Beln indicates the element of each type or range of low confidence, Bel denotes the lower limit of the propositional probability, and Disi indicates the level of distrust for each factor type or scope. Thus, if there is no landslide, the Bel value will be zero.
- More explanation is required.
Response: Agree and change made. Thank you for your good suggestion.
As an effective classification method, the FT was first considered as a multivariate tree for the promotion of decision problems [48]. Gama put forward the FT model, which combines a significant construction discriminant function with a multivariate decision tree [48,49]. Here, D is the training dataset and n is the number of samples (Xi, Yi), , . Xi is an input variable, which in this paper refers to the 16 landslide conditioning factors. Yi is an output, which can be expressed as landslide or non-landslide. FT first establishes a decision tree to separate these two classes from the training dataset [50]. The FT algorithm uses a logistic regression function to segment the internal nodes (called oblique crack) and make an estimate on the leaves. Then, P(X) is the predictive value (PV) of the measured probability and the logical enhancement of the iteratively reweighted least squares method is determined for each class Yi (for each output comprising two classes) [41,50].
where Xi is the input variable and βi is the modulus of the i-th component.
- Please provide the description of the study area before the description of methods.
Response: Agree and change made. Thank you for your good suggestion. We exchanged two chapter positions for the study area description and method description, which makes the structure of the article more reasonable.
- Change the color of elevation. Use green to red band.
Response: Agree and change made. Thank you for your good suggestion.
Figure 1. The study area.
- Try to simplify the sentences.
Response: Agree and change made. Thank you for your good suggestion. The whole manuscript has been proofread by MDPI (https://www.mdpi.com/authors/english ), you can see its certificate.
Nanchuan District is located on the southeastern margin of the Sichuan Basin and to the northwest of the Dalou Mountains. The area has the characteristics of strong settlement in the northwest (within the basin) and uplift in the southeast. The lithology in the area is mainly mudstone, sandstone, and limestone, and Quaternary deposits is widely distributed in depressions, river valleys, and slopes. Paleozoic and Mesozoic deposits are widely distributed in the area. The lithology is mainly carbonate rocks and clastic rocks, and there are a few Quaternary loose accumulation layers, which laid the foundation for the formation of groundwater and constituted carbon. The three basic types of groundwater are karst water in salt rock, fissure water in bedrock, and pore water in loose rock. The surface water system in the area mainly comprises the Yangtze River system, which is mostly branched, followed by feathers, with a large slope drop and rapid water flow.
- Large sacle of small scale or for a large area. Please clarify it.
Response: Agree and change made. Thank you for your good suggestion. The large scale here refers to the area size of the selected study area, indicating that the selected study area is large. This point was put forward by Ercanoglu, and Adineh et al. Approved this point. Thank you for giving me the opportunity to present my views.
Ercanoglu M. Landslide susceptibility assessment of SE Bartin (West Black Sea region, Turkey) by artificial neural networks. Nat. Hazards Earth Syst. Sci. 2005, 5: 979992.
Adineh, F.; Motamedvaziri, B.; Ahmadi, H.; Moeini, A. Landslide susceptibility mapping using genetic algorithm for the rule set production(garp) model. J Mt. Sci. 2018, 15, 167-180.
- Please simplify the sentence.
Response: Agree and change made. Thank you for your good suggestion. The whole manuscript has been proofread by MDPI (https://www.mdpi.com/authors/english ), you can see its certificate.
In the overall process of landslide susceptibility prediction, based on statistics, it is assumed that the conditions of future landslides are the same as those of the past [16,60].
- How it can be said as a literature search. e –“In this study, 298 landslides were identified by using a literature search. By referring to the historic reports, Google Earth satellite image interpretations, and through field investigation, landslide areas were converted into point data (landslide centroids).”
Response: What we want to express here is that the landslide we identified came from literature search, historical reports, and Google Earth satellite image interpretations, not a single literature search. Improper expression has been corrected. Thank you for your good suggestion and give me the opportunity to express my opinion. This error is sorry for your trouble.
In this study, 298 landslides were identified based on field investigations, historical reports, and Google Earth satellite image interpretations. A complete landslide inventory map of Nanchuan District was used to identify and record the location (centroid) of these previous landslides, which consists of 296 slides and 3 rockfalls (Figure 1).
- Repetation?
Response: Dear reviewer, this part is about causality. Because on the basis of some references, we chose the proportion of 7:3. Thank you for giving me the opportunity to present my views.
Furthermore, through the analysis and comparison of landslide data, it was found that 70%:30% could be used as the classification standard of landslide data in this paper [62-64]. In addition, an equal amount (298) of non-landslide points were randomly selected in areas without landslides, and then allocated 70%/30% to the training and verification data sets, respectively. Then, data were assigned values of 1 (with landslide) and 0 (without landslide).
- Simplify these sentences.
Response: Agree and change made. Thank you for your good suggestion. The whole manuscript has been proofread by MDPI (https://www.mdpi.com/authors/english ), you can see its certificate.
Compilation of landslide inventory maps requires selection and creation of landslide conditioning factors [28]. These factors are mainly selected according to three aspects: geological factors, topographic factors, and geological environment factors.
- Divide this sentences into couple of simple senetnces.
Response: Agree and change made. Thank you for your good suggestion. The whole manuscript has been proofread by MDPI (https://www.mdpi.com/authors/english ), you can see its certificate.
Based on the existing characteristics and geological conditions of the research area and literature review [65-68], 16 conditioning factors were selected for this paper: altitude, slope angle, slope aspect, plan curvature, profile curvature, sediment transport index (STI), stream power index (SPI), topographic wetness index (TWI), the normalized difference vegetation index (NDVI), land use, geological age groups, soil, distance to roads, distance to rivers, distance to faults, and rainfall (Figure 3).
- Why this resolution have been chosen?
Response: The landslide conditioning factors we selected are all for constructing the research to obtain the landslide inventory map and prepare for the landslide susceptibility assessment. The reason why they are all converted to 20 m × 20 m is to facilitate the unification of the data format and is more conducive to the prediction of landslides susceptibility. Thank you for giving me the opportunity to present my views.
- How many weather stations? Please use Kriging method instead of IDW.
Response: 112 weather stations of Chongqing Meteorological Bureau (Figure 1). Due to the confidentiality of data, it can not be mentioned in the paper. The rainfall map based on Kriging method is drawn (Figure 2) and compared with the existing rainfall map based on the distance weighted inverse method. Obviously, the rainfall map based on Kriging method is not satisfactory. Comprehensive comparison shows that Kriging method is worse than distance weighted inverse method in modeling effect and index. Therefore, we can only continue to choose the rainfall map based on the distance weighted inverse method. At the same time, the following articles all use IDW method to get the rainfall map with good quality. I hope you will agree with us. Thank you for your advice and give me the chance to express my opinion.
Mandal, B.; Mandal, S. Analytical hierarchy process (ahp) based landslide susceptibility mapping of lish river basin of eastern darjeeling himalaya, india. Adv. Space Res. 2018, 62, 3114-3132.
Hong, H.; Liu, J.; Bui, D.T.; Pradhan, B.; Acharya, T.D.; Pham, B.T.; Zhu, A.X.; Wei, C.; Ahmad, B.B. Landslide susceptibility mapping using j48 decision tree with adaboost, bagging and rotation forest ensembles in the guangchang area (china). Catena 2018, 163, 399-413.
Long N T, De Smedt F. Analysis and mapping of rainfall-induced landslide susceptibility in a Luoi District, Thua Thien Hue Province, Vietnam. Water, 2019, 11(1): 51.
|
Figure 1.Rainfall station data in the study area |
Figure 2. rainfall map based on Kriging method |
- Can you please increase the font of legend.
Response: In order to maintain the consistency of the legend in this group of pictures and the overall coordination of the picture, this is already the largest and largest font in the legend. Thank you for your good advice and give me the opportunity to express my opinion.
Response: Agree and correct. Thank you for your good advice. What we want to express here is that the thematic layer of landslide data and the thematic map of 16 conditioning factors are combined with each other, and the proportion of each category in each factor is calculated. Sorry for the trouble caused by improper expression. Thank you for giving me the opportunity to present my views.
In this paper, the thematic layer of landslide data and the thematic map of 16 conditioning factors were combined to compute the number of pixels and landslides under different categories, and the proportion of each category was calculated.
- Interdependency or independency.
Response: Agree and change made. Thank you very much for your good suggestions.
Multicollinearity can be used to test the possible interdependency between the conditioning factors of a landslide.
- Simplify this sentence.
Response: Agree and change made. Thank you for your good suggestion. The whole manuscript has been proofread by MDPI (https://www.mdpi.com/authors/english ), you can see its certificate.
Therefore, a multicollinearity analysis was carried out to determine if there is interdependence between the adjustment factors of the EBF model preprocessing.
- Please check the image.Slope angle and altitude??
Response: Agree and change made. Thank you for your good suggestion. We have not listed the specific AM and its standard deviation here, so it is difficult to understand the existence. The standard deviation of slope angle and altitude is 0, so it is not shown in the figure. Now, on the premise that the data is accurate and reliable, and the error does not affect the prediction and comparison of various factors, the upper and lower limits of the standard deviation are removed.
Figure 5. The prediction ability of the landslide conditioning factors.
- Can you please discuss the coefficients?
Response: Agree and change made. Thank you for your good suggestion.
Based on the dependent variables (landslide data) and independent variables (16 conditioning factors), the values of various conditioning factors in the LR model are calculated using the Weka software. The selected test mode is 10-fold cross-validation. Meanwhile, the linear combination equation of the LSI of the LR model was constructed (Equation (12)). The intercept (−19.0897) and the coefficient of every conditioning factor is shown in the formula. The coefficient size of each factor is different, and the contribution to LSI is also different. From these coefficients, in the LR model, slope angle and TWI have the greatest impact on LSI, while the negative NDVI and profile curvature have a suppressive effect on LSI.
- Please check the image. AUC of line 1 meaning?
Response: Here, AUC of line 1 means the line formed by the calculated AUC value composed of all parameters in the first row of Table 3. The selected parameters include numBoostingIterations iterations from -1 to 30, splitOnResiduals of false, and useAIC of false. And so on, there will be four lines in a figure. Thank you for giving me the opportunity to present my views.
- Simplify the sentence.
Response: Agree and change made. Thank you for your good suggestion. At the same time, we choose four classification methods to optimize the landslide susceptibility map, and finally determine the quantile as the classification scheme for mapping. The whole manuscript has been proofread by MDPI (https://www.mdpi.com/authors/english ), you can see its certificate.
In this study, a unique LSI was applied to all pixels in this research area to establish a landslide susceptibility model. The classification methods of LSI mainly include Jenks natural breaks, quantile, geometrical interval, equal interval, and standard deviation [92,93]. In this study, the four classification methods of Jenks natural breaks, equal interval, quantile, and geometrical interval were used to divide LSI into five categories: very low (VLC), low (LC), moderate [54], high (HC) and very high (VHC). The relative distribution of susceptibility category areas and the relative proportion of landslides in each category in the study are shown in Figure 11. Generally, the histograms of different models under different classification methods show regularity: the higher the susceptibility category, the greater the landslide distribution. Most landslides were recorded in the very high (VHC) category. It can be clearly seen from Figure 11 that this rule exists, and the quantile is superior to the other three classification methods, with outstanding performance. In the FT model, the quantile classification method does not perform as well as the other three models, but for the FT model, the quantile classification method is still its best choice. Therefore, the quantile is used as a classification scheme for landslide susceptibility maps of the EBF, FT, LR, and LMT models.
- What does it mean?
Response: This means that EBF model is easy to model, and does not need too much training process and parameter adjustment. Thank you for your good suggestion and give me the opportunity to express my opinion.
As a traditional bivariate model, the EBF model is easy to understand and operate, and does not require a complex training process or adjustment of various parameters.
- This word should not be used in a scientific paper.
Response: Agree and change made. Thank you for your good suggestion. The whole manuscript has been proofread by MDPI (https://www.mdpi.com/authors/english ), you can see its certificate.
Meanwhile, statistical analysis of the landslide dataset was conducted, and the data distribution, mean standard errors, standard deviation, and variances of each model were obtained.
In short, we have made strict content additions and format corrections to this article, including:
- Correction and position exchange of research area and landslide inventory.
- Added the content of the results section, and analyzed the distribution of landslides in the categories obtained by different classification methods.
- Major changes to the discussion.
- We checked the manuscript several times to eliminate typographical and typographical errors, including forms and numbers. And MDPI proofread the whole manuscript (https://www.mdpi.com/authors/english ).
Once again, I would like to thank you for giving me this opportunity to present our views. Hope to get your approval.
Waiting for your positive response.

Reviewer 3 Report
The manuscript entitled “Optimization of computational intelligence models for landslide susceptibility evaluation”, by X. Zhao and W. Chen, presents an improved and good work.
The manuscript should be acceptable for publication in the present form.
Author Response
Dear Editor-in-Chief
Many thanks for useful comments and suggestions on our manuscript entitled "Optimization of computational intelligence models for landslide susceptibility evaluation" (remotesensing-824363). The suggestions are quite helpful for improving the MS and we incorporate them into the revised manuscript. We have addressed all concerns as outlined below. We have made use of the language editing service provided by MDPI (with a certificate). Finally, we hope that the revised manuscript can be acceptable in its present form.
Thanking you in advance.
Yours Sincerely,
COMMENTS FROM EDITORS AND REVIEWERS
Peer Reviewer #3:
The manuscript entitled “Optimization of computational intelligence models for landslide susceptibility evaluation”, by X. Zhao and W. Chen, presents an improved and good work.
The manuscript should be acceptable for publication in the present form.
Response: Dear Reviewer #3, thank you so much for your positive opinions on our manuscript.

Round 2
Reviewer 1 Report
General comments:
Dear Authors,
the paper underwent substantial reorganization and integration. Now discussion and conclusions are more adequately linked to / supported by the previous sections.
I suggest some further revisions, as described in the specific comments given in the following.
Best regards.
Specific comments:
PAG. 6 “… study, 298 landslides were identified based on field investigations, historical reports, and Google Earth satellite image interpretations. A complete landslide inventory map of Nanchuan District was used to identify and record the location (centroid) of these previous landslides, which consists of 296 slides and 3 rockfalls (Figure 1).”
296 slides plus 3 rockfalls gives a total of 299 landslides. Please check.
PAG. 6 “This study uses …… landslides were less than 100,000 m3.”
Thanks to the new descriptions and information the Authors included in the text, now the readers may be aware that the inventory includes slides ranging within a very wide size interval, as well as some falls. Very hardly factors controlling/triggering landslides of hundreds m2 size (probably soil slips???) may be the same of landslides of ca. 10E4 m2, the latter most probably involving the bedrock. Could authors check if and how final accuracy results can be modified/improved by selecting as training/check subsets, landslide groups made up of slides that are quite homogeneous in terms of size? To this aim, could authors add a diagram describing the frequency distribution of slides area? Obviously, this would imply a bit of new work to be performed.
FIG. 3 d) and e)
I can understand the motivations given by the Authors, but the figures are anyway meaningless. I suggest to add two images zooming a subset area at a larger scale in order to make the general images meaningful.
Author Response
Dear Editor-in-Chief
Many thanks for useful comments and suggestions on our manuscript entitled "Optimization of computational intelligence models for landslide susceptibility evaluation" (remotesensing-824363). The suggestions are quite helpful for improving the MS and we incorporate them into the revised manuscript. We have addressed all concerns as outlined below. We have made use of the language editing service provided by MDPI (with a certificate). Finally, we hope that the revised manuscript can be acceptable in its present form.
Thanking you in advance.
Yours Sincerely,
List of changes in the revised paper:
This document explains the changes made in the revised manuscript while dealing with the comments raised by the reviewers. Reviewer’s comments are marked in black; author’s response is shown in blue, while the changes in the manuscript are marked in red.
COMMENTS FROM EDITORS AND REVIEWERS
Peer Reviewer #1:
General comments:
Dear Authors,
the paper underwent substantial reorganization and integration. Now discussion and conclusions are more adequately linked to / supported by the previous sections.
I suggest some further revisions, as described in the specific comments given in the following.
Best regards.
Response: Dear Reviewer #1, thank you so much for your positive opinions on our manuscript. We have made revisions to the paper based on your suggestions and responded to your comments point-by-point. Thank you for giving us this opportunity to introduce our views on this paper.
Specific comments:
PAG. 6 “… study, 298 landslides were identified based on field investigations, historical reports, and Google Earth satellite image interpretations. A complete landslide inventory map of Nanchuan District was used to identify and record the location (centroid) of these previous landslides, which consists of 296 slides and 3 rockfalls (Figure 1).”
296 slides plus 3 rockfalls gives a total of 299 landslides. Please check.
Response: Agree and change made. Thank you for your good suggestion.
A complete landslide inventory map of Nanchuan District was used to identify and record the location (centroid) of these previous landslides (Figure 1), which consists of 295 slides and 3 rockfalls [61].
PAG. 6 “This study uses …… landslides were less than 100,000 m3.”
Thanks to the new descriptions and information the Authors included in the text, now the readers may be aware that the inventory includes slides ranging within a very wide size interval, as well as some falls. Very hardly factors controlling/triggering landslides of hundreds m2 size (probably soil slips???) may be the same of landslides of ca. 10E4 m2, the latter most probably involving the bedrock. Could authors check if and how final accuracy results can be modified/improved by selecting as training/check subsets, landslide groups made up of slides that are quite homogeneous in terms of size? To this aim, could authors add a diagram describing the frequency distribution of slides area? Obviously, this would imply a bit of new work to be performed.
Response: Agree and change made. Thank you for your good suggestion. Yes, there is a wide range of landslides in the study area (as shown in Figure 1). The landslides studied in this paper are all found so far, all of which have different sizes. There is no choice work because of this. In addition, it is considered that the selection of landslide data is too disturbing and easy to be subjective, while the random classification by the computer can objectively measure the reliability of the inferred value in the way of probability when the overall landslide data is known. It is also a popular method to establish landslide database by the location (centroid) of each landslide. We should not ignore every landslide to carry out this landslide susceptibility prediction, so as to ensure the reliability and accuracy of the results. We calculated 298 landslides and drew two pie charts which have been inserted into the paper. Hope to get your affirmation.
According to the analysis of landslides using GIS tools, the volumes of the three rockfalls are 4,800m3, 12,000 m3 and 13,100 m3, respectively. The size distribution of 295 slides is shown in Figure 4. For slides, the smallest area was close to 70 m2, the largest was more than 8.4×105 m2, and the average area was about 3.07×104 m2. In terms of volume, the minimum volume was only 140 m3, and more than 95% of the slides were less than 100,000 m3.
|
Unit: m2 |
Figure 4. The frequency distribution of 295 slides.
FIG. 3 d) and e)
I can understand the motivations given by the Authors, but the figures are anyway meaningless. I suggest to add two images zooming a subset area at a larger scale in order to make the general images meaningful.
Response: Agree and change made. Thank you for your good suggestion.
Figure 4. Landslide conditioning factors: (d) Plan curvature, (e) Profile curvature.
Again, thank you for your good advice and give me the opportunity to present our views.
Waiting for your positive response.

Reviewer 2 Report
1. Abstract: replace (70/30) with (70%: number of landslides; 30%:.......)
2. Abstract: Remove the word pixel-based value and rewrite it.
3. ROC was used to validate and compare. It is not possible to analyze using this curve.
4. How did you define a good result? Please provide a category.
5. Please delete the line " Furthermore, the
models produced using these four approaches can provide a reference for other scholars."
6. Description of the study area: can you provide different references for the seasonal distribution of atmospheric precipitation?
7. Figure 1: Please remove _ from landslide training and validation.
8. Figure 2: remove / from 70% and put :....
9. Data Preparation: Obtaining and constructing datasets of the landslide conditioning factors and the landslide
inventory is necessary for landslide susceptibility zoning and analysis [58].
Rewrite this sentence. Obtaining and construction of data sets suit for landslide conditioning factors but do not suit for landslide inventory.
10. [59] Check the font, please.
11. . In the
the overall process of landslide susceptibility prediction, based on statistics, it is assumed that the
conditions of future landslides are the same as those of the past [16,60].
Please check the grammar of this sentence.
12. In one sentence, 70%:30% was written, and later 70%/30% was written. Please use 70%:30%.
13. Compilation of landslide inventory maps requires selection and creation of landslide
conditioning factors: What is the meaning of this sentence. Please clarify.
14. None landslides are found when the slope angle is >70°: please check this sentence.
15. Please increase the resolution of Figure 4.
16. Model Configuration: please delete the sentence: After the correlation analysis, multicollinearity test, and research on the prediction ability of the
conditioning factors, training, and verification models must be established.
17. Landslide spatial prediction, as a popular global research topic, is an important issue in land use
and management [: Please dod not use a very bombastic word.
18. Discussion: In order to achieve and in order to ensure In order to undertake... Please get rid of one of them.
19. The conclusion is too short. Please discuss the essential findings and limitations of this study.
Author Response
Dear Editor-in-Chief
Many thanks for useful comments and suggestions on our manuscript entitled "Optimization of computational intelligence models for landslide susceptibility evaluation" (remotesensing-824363). The suggestions are quite helpful for improving the MS and we incorporate them into the revised manuscript. We have addressed all concerns as outlined below. Finally, we hope that the revised manuscript can be acceptable in its present form.
Thanking you in advance.
Yours Sincerely,
List of changes in the revised paper:
This document explains the changes made in the revised manuscript while dealing with the comments raised by the reviewers. Reviewer’s comments are marked in black; author’s response is shown in blue, while the changes in the manuscript are marked in red.
COMMENTS FROM EDITORS AND REVIEWERS
Peer Reviewer #2:
1. Abstract: replace (70/30) with (70%: number of landslides; 30%:.......)
Response: Agree and change made. Thank you for your good suggestion. We have changed the original text to (70%: 209; 30%: 89).
2. Abstract: Remove the word pixel-based value and rewrite it.
Response: Agree and change made. Thank you for your good suggestion.
Then, based on the EBF method, the Bel values of 16 conditioning factors related to landslide occurrence were calculated, and these Bel values were used as input data for building other models.
3. ROC was used to validate and compare. It is not possible to analyze using this curve.
Response: Agree and change made. Thank you for your good suggestion. Here, what we want to express is to verify and compare each model according to ROC curve and AUC value. As we all know, ROC curve has become a popular method for accuracy evaluation because of its comprehensive, easy to understand and visually attractive style. This method uses the AUC for quantitative assessment, which plots 1-specificity on the x-axis against sensitivity on the y-axis. Thank you for giving me the opportunity to present my views.
The receiver operating characteristic (ROC) curve and the values of the area under the ROC (AUC) were used to evaluate and compare the prediction ability of the four models.
4. How did you define a good result? Please provide a category.
Response: The ROC curve is a coordinate graph and a high-quality tool for probability prediction systems [90,91]. In the coordinate system, the closer the point of the ROC curve to the upper left corner, the higher the accuracy of the test results. The AUC value range is [0.5, 1.0], in which the highest AUC has the best diagnostic value [92]. Then, the AUC value produced by a excellent model is between 0.9-1, good model (0.8-0.9), fair model (0.7-0.8), a poor model in the range of 0.6-0.7 and the final 0.5-0.6 poor accuracy range of the model [93]. Thank you for giving me the opportunity to present my views.
5. Please delete the line " Furthermore, the models produced using these four approaches can provide a reference for other scholars."
Response: Agree and change made. Thank you for your good suggestion.
6. Description of the study area: can you provide different references for the seasonal distribution of atmospheric precipitation?
Response: Thank you for your good suggestion. Figure 2 shows the average monthly rainfall from 1990 to 2019. In order to search for more relevant literature support on the seasonal distribution of atmospheric precipitation, Wei and Hu put forward similar views, Zhou and Li also put forward similar views on Beibei area of Chongqing, which is close to Nanchuan area.
Wei, T.; H, J. Analysis of Influences on Environment of Chongqing Wansheng-Nanchuan Expressway and Countermeasures for Environmental Protection. Technology of Highway and Transport, 2009, 6, 38.
Zhou, J.; Li, T. A tentative study of the relationship between annual δ18O & δD variations of precipitation and atmospheric circulations—A case from Southwest China. Quat. Int. 2018, 479: 117-127.
Figure 2. Average monthly rainfall from 1990 to 2019.
7. Figure 1: Please remove _ from landslide training and validation.
Response: Agree and change made. Thank you for your good suggestion.
Figure 1. The study area.
8. Figure 2: remove / from 70% and put :....
Response: Agree and change made. Thank you for your good suggestion.
Figure 3. The flowchart used in this research.
9. Data Preparation: Obtaining and constructing datasets of the landslide conditioning factors and the landslide inventory is necessary for landslide susceptibility zoning and analysis [58].
Rewrite this sentence. Obtaining and construction of data sets suit for landslide conditioning factors but do not suit for landslide inventory.
Response: Agree and change made. We deleted this sentence in order to avoid misunderstanding for future readers. Thank you for your good suggestion.
10. [59] Check the font, please.
Response: Agree and change made. Thank you for your good suggestion.
11. In the overall process of landslide susceptibility prediction, based on statistics, it is assumed that the conditions of future landslides are the same as those of the past [16,60].
Please check the grammar of this sentence.
Response: Agree and change made. We deleted this sentence in order to avoid misunderstanding for future readers. Thank you for your good suggestion.
12. In one sentence, 70%:30% was written, and later 70%/30% was written. Please use 70%:30%.
Response: Agree and change made. Thank you for your good suggestion.
13. Compilation of landslide inventory maps requires selection and creation of landslide
conditioning factors: What is the meaning of this sentence. Please clarify.
Response: Agree and change made. Thank you for your good suggestion.
Here it should be expressed as:
After compilation of the landslide inventory map, it is necessary to select and create landslide conditioning factors for landslides susceptibility prediction [28].
14. None landslides are found when the slope angle is >70°: please check this sentence.
Response: Agree and change made. we deleted this sentence in order to avoid misunderstanding for future readers. Thank you for your good suggestion.
15. Please increase the resolution of Figure 4.
Response: Agree and change made. Thank you for your good suggestion.
Figure 4. Relationship between landslides and the landslide conditioning factors of (a) Altitude, (b) Slope angle, (c) Slope aspect, (d) Plan curvature, (e) Profile curvature, (f) STI, (g) SPI, (h) TWI, (i) NDVI, (j) Landuse, (k) Geological age groups, (l) Soil, (m) Distance to roads, (n) Distance to rivers, (o) Distance to faults, (p) Rainfall.
16. Model Configuration: please delete the sentence: After the correlation analysis, multicollinearity test, and research on the prediction ability of the conditioning factors, training, and verification models must be established.
Response: Agree and change made. We have deleted the sentence in the revised manuscript. Thank you for your good suggestion.
17. Landslide spatial prediction, as a popular global research topic, is an important issue in land use and management [: Please dod not use a very bombastic word.
Response: Agree and change made. Thank you for your good suggestion.
Landslide spatial prediction is an important issue in land use and management.
18. Discussion: In order to achieve and in order to ensure In order to undertake... Please get rid of one of them.
Response: Agree and change made. Thank you for your good suggestion.
Analyzing the performance of the model and optimizing the preform model can ensure the quality of the modeling, establish a robust landslide inventory map, and select 16 landslide conditioning factors from it.
In this study, the selected models are optimized, the optimal parameters are determined, and excellent parameter configuration is used to ensure the optimization of the model, so as to obtain better model performance.
19. The conclusion is too short. Please discuss the essential findings and limitations of this study.
Response: Agree and change made. Thank you for your good suggestion.
Generally, landslide susceptibility zoning is a useful tool for landslide disaster management and planning. This research was based on four different algorithms (EBF, FT, LR, and LMT) for landslide susceptibility spatial prediction in Nanchuan District. The conditioning factors, landslide datasets, and models were analyzed and improved to achieve the most suitable algorithm. The main results are summarized as follows:
(1) The maps showed that the four landslide susceptibility models were adequate for landslide susceptibility zoning. Compared with the EBF, LR, and FT models, the LMT model showed the best performance.
(2) According to the results of the EBF model, most landslides occur at altitudes of 900-1100m in the southwest, with a slope angle of 10-20°, plan curvature of −0.05 to 0.05, profile curvature of −27.51 to −0.05, STI >20, SPI> 20, TWI of 0.24–1, NDVI of 0.20–0.26, farmland category in landuse, the fifth group (Ordovician: greyish-black charcoal shale, siliceous base) in lithology, Dystric Cambisol category in soil, 0–200 m distance to roads, 200–400 m distance to rivers, 1000–2000 m distance to faults, 333.62–1221.86 category in rainfall.
(3) According to the results of the attribute evaluation method, the most factors influencing the occurrence of landslide were the altitude, slope angle, slope aspect, plan curvature, profile curvature, STI, SPI, TWI, NDVI, landuse, geological age groups, soil, distance to roads, distance to rivers, distance to faults, and rainfall.
(4) The landslide susceptibility mapping by quantile classification scheme can be a promising tool for government decision makers and engineering technicians.
This method successfully compared four different models and explored the landslide sensitivity in Nanchuan District. The developed method can be used in landslide management and land planning. However, in the future, the ensemble of machine learning techniques for landslide susceptibility modeling still needs to be tested in different cases.
Again, thank you for your good advice and give me the opportunity to present our views.
Waiting for your positive response.

This manuscript is a resubmission of an earlier submission. The following is a list of the peer review reports and author responses from that submission.
Round 1
Reviewer 1 Report
Dear Authors,
The manuscript does not deal with the topics usually developed by this Journal because no remote sensing techniques are implemented and discussed, and the objectives do not deal with remote sensing topics also.
English refuses and errors are widespread in the text.
Regarding the contents, the topic involved is interesting. Anyway, the literature review within the introduction should be largely improved. The methods description is good enough, but there is too few information about the adopted models. The Authors are also invited to improve the description of the landslide inventory. As an example, at least landslide types as well as frequency-area distribution are required information in order to characterize the features populating the inventory. Discussion is in my opinion too “superficial” and, in more detail, the analysis of the accuracy of the results should be developed and treated in more detail. This is very important when comparing results among different models.
More specific comments are given in the following.
Best regards.
Specific comments:
Pag.1 “Landslide susceptibility mapping is one indispensable tool for landslide control [6]. The selection of appropriate analysis, the choose of modeling methods, the scale of work, and the quantity and quality of available data determine the reliability of landslide susceptibility maps [7].”:
Authors should improve literature citations, there is a lot of specific literature about these topics
Pag. 2 “However, because of its high cost, it is not suitable for large areas. Traditional statistical analysis requires the presumption of an appropriate structural model. It then focuses on parameterization, which is widely applied to the analysis of geological disasters, such as landslides [15,17,18]. However, this method is not practical because of its underlying hypothesis and the over-dependence on parameters [15], such as the certainty factor [19,20], Dempster-Shafer [21,22], entropy [19,23], and weight of evidence [24,25].”
Wider description of statistical models is necessary. Also, the meaning of "appropriate structural model" is unclear to the reader and should be better clarified. The list of statistical methods is located in a “non-sense” position within the sentence.
Pag. 2 “These methods overcome the shortcomings of physically-based and statistically-based methods, and produce more reliable results through data statistics [28,29]. Many machine learning algorithms have been applied, such as random forest [30,31], artificial neural network [32-34], support vector machine [35,36], and naive Bayes classifiers [37,38].
Goetz at al. (2015) stress that there are not-significant differences between the models they used, actually. Model names should have capital letters (Artificial Neural Network). All the paper should be reviewed about this task. As an example, in Reichnebach et al. (2018) a well-developed review of data-driven methods may be found.
Pag. 5-6 “Study Area”
A description of the geological setting of the area is completely missing. It would be useful to use a map instead of describing the drainage network. Moreover, the terms used to describe the drainage network appear to be are inappropriate and more strict reference to the relevant literature should be done for specific terms.
Pag. 7 “Ercanoglu proposed that obtaining and constructing datasets of the landslide inventory and the landslide conditioning factors are necessary for large-scale landslide susceptibility zoning and analysis [54]. The location of previous landslides can be determined in the landslide inventory map.”
It seems there is no reference to this Authors in the bibliography. Moreover, this sentence should be moved in the Introduction chapter. The sentence “location of previous landslides can be determined in the landslide inventory map” is not clear to the reader: a landslide inventory IS a database recording occurred landslides; without a landslide inventory map you are not able to know the location of previous landslides. The Authors should better develop this section.
Pag. 7 “In this study, 298 landslides were identified by collecting information on landslides that occurred in the region, as provided in early reports and literature. By referring to the Google Earth satellite image and Landsat 8 interpretations, and through field investigation, pixel-based landslide areas were converted into point data (landslide centroids).”
Again, the general picture is rather unclear for the reader. Some questions arise: Did the entire inventory exist before performing this study? Which are the sources for the reports and related literature? What about accuracy of results of remote imagery interpretation?
Pag. 7 “Digital elevation model (DEM) was extracted from Aster GDEM data (http://www.gscloud.cn). DEM were used to extract thematic data layers such as elevation, slope angle, aspect, plane curvature, section curvature, STI, SPI and TWI.1:200,000 geological maps, 1:50,000 scale topographic map, and landsat 8 OLI images were used to extract the NDVI, distance to roads, distance to rivers, and distance to faults. Using the local rainfall data, a rainfall map was made. The landuse thematic data were extracted from the 1:100,000 landuse map. Meanwhile, the soil thematic data were made from the 1:1,000,000 scale soil map.”
Geological map, land use map and soil map references should be reported. Please, a) also the code/software used to obtain the raster outputs from the DEM should be reported; b) the map of rainfall gauges (chapter study area) should be added; c) the method used to interpolate the rainfall data should be described.
Figure 3: Please, select different colour patterns for curvatures. However, at this scale, the curvatures maps seem to not discriminate morphology.
Pag. 10 “Regarding the slope angle, the EBF value of the southwest slope (0.148) is the largest, and that of the plane is zero”
Unclear to the reader. Do you mean slope aspect?
Fig. 4: The authors should improve the quality of graphs and insert the axes titles.
Pag. 12 “Multicollinearity analysis”
The authors should improve both the description of the multicollinearity analysis methods and interpretation of the results.
Pag. 18 “In this paper, when finding the adjacent features with relatively large differences in the LSI values, the LSI values were divided into five categories and the natural break method was adopted, including very low, low, medium, high, and very high, which were transformed into ArcGIS to generate the ultimate map.”
Natural breaks approach is an intuitive and fast method that classifies on the basis of the values distribution. However, each susceptibility model will have its own distribution of values depending on the algorithm used to make calculations. Since the goal of the work is to perform comparison among different methods, it is necessary to use one or more classification methods that may be adequate toi describe all the models output (as an example: quantiles).
Pag. 20 “Discussion”
This chapter is poor, and the topics of the paper are not adequately explored. As an example, it would be interesting for the readers to understand the distribution of landslides within classes of the different approaches. The Authors should also explain how to interpret the results shown within "Figure 11".
Reviewer 2 Report
This paper has some merits but has several issues. I have attached my comments, suggestions, and correction in the attached doc file. Major issues are:
1) The structure of the paper is not correct. Please check the attached file.
2) There are several fundamental issues. Please check the attached file.
3) writing has to be improved. Please check the attached file.
4) I would advise authors to read some papers on a landslide to correct the use of terminologies related to landslide susceptibility mapping.

Reviewer 3 Report
The manuscript entitled “Optimization of computational intelligence models for spatial prediction of landslide susceptibility”, by X. Zhao & W. Chen, presents an interesting work.
In general, the manuscript should be acceptable for publication but some serious problems must be repaired prior to publication. It needs some significant improvement. Some suggestions are as follows:
- Please use different terms in the “Title” and the “Keywords”.
- You could enrich the conclusions.
- It would be useful to be described the aim of this paper.
- The English language usage should be checked by a fluent English speaker. It is suggested to the authors to take the assistance of someone with English as mother tongue.
- You could enrich the scientific literature.
- Please justify convincingly why this manuscript (method, thematology etc) connected with Remote Sensing’s content and scope. Perhaps the using of proper literature from the journal would be helpful. Eg:
- Lee, D.-H.; Kim, Y.-T.; Lee, S.-R. Shallow Landslide Susceptibility Models Based on Artificial Neural Networks Considering the Factor Selection Method and Various Non-Linear Activation Functions. Remote Sens. 2020, 12, 1194.
- Chang, Z.; Du, Z.; Zhang, F.; Huang, F.; Chen, J.; Li, W.; Guo, Z. Landslide Susceptibility Prediction Based on Remote Sensing Images and GIS: Comparisons of Supervised and Unsupervised Machine Learning Models. Remote Sens. 2020, 12, 502.
- Arabameri, A.; Pradhan, B.; Rezaei, K.; Lee, C.-W. Assessment of Landslide Susceptibility Using Statistical- and Artificial Intelligence-Based FR–RF Integrated Model and Multiresolution DEMs. Remote Sens. 2019, 11, 999.
- etc, etc.
7. The authors could make a discussion about the relationship between the landslide susceptibility and planning. See the following publications:
- Skilodimou, H.D.; Bathrellos, G.D.; Koskeridou, E.; Soukis, K.; Rozos, D. Physical and Anthropogenic Factors Related to Landslide Activity in the Northern Peloponnese, Greece. Land 2018, 7, 85.
8. Correct references in the text and the reference list according to the journal’s format. Please format the references’ list by using the correct journal abbreviations.
See the following link: https://images.webofknowledge.com/images/help/WOS/A_abrvjt.html
9. In which journals are published the references 33, 44, 73 & 74? Please fix it.